# Initialization is Critical to Whether Transformers Fit Composite Functions by Reasoning or Memorizing

**Zhongwang Zhang**[1,2], **Pengxiao Lin**[1,2], **Zhiwei Wang**[1,2], **Yaoyu Zhang**[1,2], **Zhi-Qin John Xu**[1,2,3,4,5*]

[1] Institute of Natural Sciences, MOE-LSC, Shanghai Jiao Tong University
[2] School of Mathematical Sciences, Shanghai Jiao Tong University
[3] Key Laboratory of Marine Intelligent Equipment and System, Ministry of Education, P.R. China
[4] Shanghai Seres Information Technology Co., Ltd, Shanghai, P.R. China
[5] Center for LLM, Institute for Advanced Algorithms Research, Shanghai

## Abstract

Transformers have shown impressive capabilities across various tasks, but their performance on compositional problems remains a topic of debate. In this work, we investigate the mechanisms of how transformers behave on unseen compositional tasks. We discover that the parameter initialization scale plays a critical role in determining whether the model learns inferential (reasoning-based) solutions, which capture the underlying compositional primitives, or symmetric (memory-based) solutions, which simply memorize mappings without understanding the compositional structure. By analyzing the information flow and vector representations within the model, we reveal the distinct mechanisms underlying these solution types. We further find that inferential (reasoning-based) solutions exhibit low complexity bias, which we hypothesize is a key factor enabling them to learn individual mappings for single anchors. We validate our conclusions on various real-world datasets. Our findings provide valuable insights into the role of initialization scale in tuning the reasoning and memorizing ability and we propose the initialization rate $\gamma$ to be a convenient tunable hyper-parameter in common deep learning frameworks, where $1/d_{\text{in}}^{\gamma}$ is the standard deviation of parameters of the layer with $d_{\text{in}}$ input neurons.

## 1   Introduction

Large-scale transformers (Vaswani et al., 2017) demonstrate unprecedented capabilities (Achiam et al., 2023; Brown et al., 2020; Choi et al., 2021; Teubner et al., 2023), even noted as "sparks of AGI" (Bubeck et al., 2023). However, their performance on compositional tasks, which are considered a key property of human cognition (Marcus, 2003; Smolensky et al., 2022), has long been the subject of intense debate, especially in terms of the out-of-distribution (OOD) generalization ability. This raises critical open questions about how to faithfully interpret transformers' capabilities on compositional tasks. Can transformers learn the underlying compositional primitives within the data, or do they merely learn input-output mappings? When transformers produce incorrect outputs on compositional tasks, do their responses follow any discernible patterns or logic?

In this work, we use anchor functions (Zhang et al., 2024), a framework for creating controlled synthetic data, to investigate how transformers generalize to unseen compositional tasks. In our setup, each sequence contains an anchor pair, a key item, and noise items unrelated to the output (right side of Fig. 1(a)). The single anchors are specific tokens (i.e., 1, 2, 3, 4), each corresponding to a particular arithmetic operation (left side of Fig. 1(a)). A composite function is defined as the

---

*Corresponding author: xuzhiqin@sjtu.edu.cn

38th Conference on Neural Information Processing Systems (NeurIPS 2024).

sequential application of two single anchor functions, i.e., the corresponding arithmetic operation, to a key item. For example, given a key item $x$ and an anchor pair $(1, 2)$, the composite function would output $(x + 5) + 1$, i.e., $x + 6$. As shown in the middle part of Fig. 1(a), during training, 14 out of the 16 possible anchor pairs are assigned inferential (reasoning-based) mappings, meaning the target output is consistent with the result of applying the composite function. One pair $(3, 4)$ is assigned a non-inferential (memory-based) mapping, where the target output deviates from the result of the composite function. The remaining pair $(4, 3)$ is held out as an unseen task. After training, there are three possible solutions the model could learn for the unseen pair $(4, 3)$: a symmetric solution matching the non-inferential seen pair $(3, 4)$ (Mechanism 1 in Fig. 1(b)), an inferential solution (Mechanism 2 in Fig. 1(b)), or other non-inferential solutions. The central question we aim to answer is: *which of these three types of solutions will the model learn for the unseen anchor pair (4, 3)?*

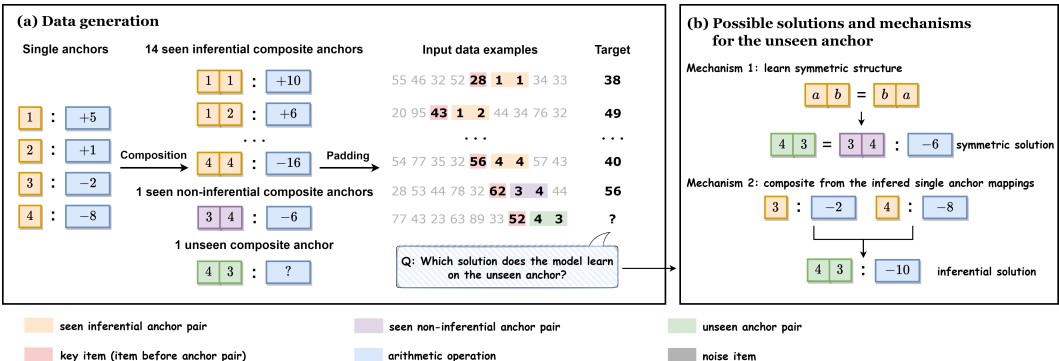

Figure 1: Experimental setup and possible solutions and mechanisms for the unseen anchor pair (4, 3). (a) Data generation: Left: The single anchors (i.e., 1, 2, 3, 4) correspond to specific arithmetic operations. Middle: During training, 14 out of the 16 possible anchor pairs are assigned inferential mappings, one pair (3, 4) is assigned a non-inferential mapping, and the remaining pair (4, 3) is held out as an unseen task (does not appear in the training). Right: The input sequences comprise an anchor pair, a key item preceding the anchor pair, and noise items unrelated to the target. The question mark indicates the output for the unseen anchor pair (4, 3), which depends on the learned solution. (b) Two potential mechanisms for the unseen anchor pair (4, 3): learning the symmetric structure (Mechanism 1) or composing the inferred single anchor mappings (Mechanism 2).

Based on the above setup, we find that the model exhibits two distinct phases of solutions depending on the parameter initialization scale with different model depths. With small initialization scales, the model tends to learn inferential solutions on the unseen anchor pair (4, 3). In contrast, with larger initialization scales, the model is more likely to learn symmetric solutions on anchor pair (4, 3) that simply match the output of its symmetric anchor pair (3, 4). These findings suggest that the initialization scale plays a crucial role in determining the type of solution learned by the model.

To gain insights into the mechanisms underlying these different solution types, we analyze the information flow and vector representations within the model. We find that symmetric solutions directly combine anchor information without capturing the underlying compositional primitives, while inferential solutions learn individual mappings for each single anchor and compose them to generate the final output. We hypothesize that the model's ability to learn compositional primitives is influenced by its complexity, determined by the initialization scale. Inferential solutions exhibit low complexity, with weights condensing in a few directions and input tokens arranged according to their numerical value in the embedding space, while symmetric solutions show no obvious condensation and lack clear structural features, indicating higher complexity.

Our work highlights the crucial role of initialization scale in shaping the solutions learned by transformers on compositional tasks, enabling them to better capture compositional structures and generalize to unseen tasks. Furthermore, adapting the initialization scale to the specific task type could be promising: larger scales for memorization tasks such as text memorization, and smaller scales for reasoning tasks such as code generation. Unless otherwise specified, our main text primarily focuses on a single-head attention model to facilitate a clearer understanding of the underlying mechanisms. We further extend our experiments to GPT-2 (Radford et al., 2019), and verify that the

insights drawn from the single-head model remain valid for more complex architectures on various real-world datasets.

With the support of the theoretical works and extensive experiments, we propose the initialization rate $\gamma$ to be a convenient tunable hyper-parameter in common deep learning frameworks, where $1/d_{\text{in}}^{\gamma}$ is the standard deviation of parameters of the layer with $d_{\text{in}}$ input neurons.

## 2 Related Work

**Challenges of transformers in compositional tasks.** Recent advancements in large language models (LLMs) have showcased remarkable capabilities, often surpassing human performance (Fu et al., 2022; Wei et al., 2022). However, despite their impressive performance on single-step reasoning tasks (Srivastava et al., 2022), transformers struggle with multi-step compositional tasks and OOD generalization (Csordás et al., 2021, 2022; Dziri et al., 2024; Hupkes et al., 2018; Lepori et al., 2023; Okawa et al., 2023; Yun et al., 2022). Ramesh et al. (Ramesh et al., 2023) show that transformers trained to directly compose capabilities struggle to generalize to OOD tasks with a synthetic task. Liu et al. (Liu et al., 2022) suggest that shallow transformers learn shortcuts during training, leading to poor OOD generalization. Numerous studies have explored various approaches to address these limitations, such as encouraging explicit reasoning step generation within a single generation (Wei et al., 2022), leveraging LLMs to generate reasoning steps iteratively (Creswell et al., 2022; Creswell and Shanahan, 2022). Despite these efforts, achieving complete mastery in compositional tasks remains a significant challenge for vanilla transformers. A series of works studies the internal mechanisms of language models and improves the capabilities of language models (Wang et al., 2024a,b, 2023; Cao et al., 2024). In order to clearly study the behaviors and internal mechanisms of language models, Zhang et al. Zhang et al. (2024) introduced anchor functions as benchmark functions for investigating transformer behavior. Our work builds on the anchor function setting to explore how different initialization scales affect model solutions and mechanisms.

**Parameter initialization of neural networks.** The parameter initialization of the network is important to determine the final fitting result of the network (Arora et al., 2019; Chizat and Bach, 2018; Zhang et al., 2019; E et al., 2020; Jacot et al., 2018; Mei et al., 2018; Rotskoff and Vanden-Eijnden, 2018; Sirignano and Spiliopoulos, 2020; Williams et al., 2019). Luo et al. (Luo et al., 2021), Zhou et al. (Zhou et al., 2022) mainly identify the linear regime and the condensed regime for two-layer and three-layer wide ReLU NNs, respectively. A series of works suggests that the condensed networks are often accompanied by good generalization ability of the model (Zhang et al., 2022; Zhang and Xu, 2023; Zhang et al., 2023; Zhang and Xu, 2024). A line of works links the grokking phenomenon with the improvement of reasoning ability (Power et al., 2022; Wang et al., 2024; Gopalani et al., 2024) and points out that the initialization scale has an important influence on the occurrence of grokking (Liu et al., 2022). Recent studies also investigate the impact of initialization on the training process of LLMs (Huang et al., 2020; Liu et al., 2020; Trockman and Kolter, 2023; Wang et al., 2024; Zhang et al., 2019; Zhu et al., 2021). These works primarily focus on how the initialization scale affects the stability of the training process and plays a crucial role in ensuring smooth and effective training of LLMs. In our work, we find that initialization scales can significantly influence a model's ability to memorize and infer data on the compositional task, highlighting the profound impact of initialization on the final performance and underlying mechanisms of the trained models.

## 3 Definitions

We introduce a set of key definitions that will be used throughout the paper. For detailed explanations and illustrative examples to better understand these definitions, please refer to Appendix B.

### 3.1 Two-anchor composite function

A two-anchor composite function $f(X) : \mathbb{R}^n \to \mathbb{R}$ is defined as

$$f(x_1, \ldots, x_n) = g\left(g(x_{i-1}; x_i); x_{i+1}\right), \quad \text{where} \quad x_i, x_{i+1} \in A. \tag{1}$$

Here, the input sequence $X = (x_1, \ldots, x_n)$ comprises $n$ tokens. An anchor set $A = \{a_1, a_2, \ldots, a_J\}$ is designated, with each token $a_k \in A$ corresponds to a function $g(x; a_k)$. In each $X$, one and only one pair of two consecutive elements belong to $A$, such as $x_i, x_{i+1} \in A$.

We refer to the token immediately preceding the anchor pair as the key item. To simplify notation, we denote the two-anchor composite function as $f(x_{i-1}; x_i, x_{i+1})$ to emphasize the anchor pair $(x_i, x_{i+1})$ and key item $x_{i-1}$.

In this work, we set the anchor set $A = \{1, 2, 3, 4\}$. Each anchor token corresponds to a specific function:

$$g(x; 1) = x + 5, \quad g(x; 2) = x + 1, \quad g(x; 3) = x - 2, \quad g(x; 4) = x - 8. \tag{2}$$

## 3.2 Data Generation

In this work, we construct the input dataset using four anchors (i.e., 1, 2, 3, 4) and items sampled from 20-99. Each sequence includes an anchor pair, a key item (the item immediately preceding the anchor pair), and noise items. The noise items are unrelated to the target. The four anchors form 16 anchor pairs, and we select a subset or all of these pairs to construct the training dataset based on the task requirements.

By default, the target is the output of the input data processed by the two-anchor composite function, i.e., the inferential mapping defined below. We divide the training data and test data based on the value of the key item. The specific division method can be found in Appendix C.

## 3.3 Mapping Type of an anchor pair

For an anchor pair $(a_1, a_2)$, the type of its mapping $\mathcal{M}_{(a_1, a_2)}(\cdot)$ can be classified as follows.

**Inferential mapping.** The designated target mapping of an anchor pair $(a_1, a_2)$ is consistent with the two-anchor composite function, i.e., $\mathcal{M}_{(a_1, a_2)}(x) = f(x; a_1, a_2)$. This type of solutions exemplifies the reasoning ability of models.

**Non-inferential mapping.** The designated target mapping of an anchor pair $(a_1, a_2)$ is inconsistent with the two-anchor composite function, i.e., $\mathcal{M}_{(a_1, a_2)}(x) \neq f(x; a_1, a_2)$.

**Symmetric mapping.** The designated target mapping of an anchor pair $(a_1, a_2)$ is consistent with the mapping of its symmetric anchor pair $(a_2, a_1)$, i.e., $\mathcal{M}_{(a_1, a_2)}(x) = \mathcal{M}_{(a_2, a_1)}(x)$.

We refer to a model as an **inferential (non-inferential, symmetric) solution** on an anchor pair if it tends to output the mapping corresponding to the inferential (non-inferential, symmetric) mapping for the studied pair.

## 3.4 Generalization

The division of the dataset naturally leads to the following two concepts of generalization:

**Generalization on data.** Generalization on data relies on the test set (defined in Appendix C). In this test data, all anchor pairs (i.e., the task) are seen in the training set.

**Generalization on task.** Generalization on task depends on the unseen anchors, i.e., anchors that do not appear in the training set, with a designated target mapping.

## 3.5 Model Architecture and Basic Experimental Setups

To enable a more focused analysis of the underlying mechanisms, the following sections will only introduce the architecture of the single-head attention model.

The input sequence is represented as a one-hot vector $X^{\mathrm{in}}$. The word embedding $X^{\mathrm{em}}$ and the input to the first transformer block $X^{(1)}$ is calculated as:

$$X^{\mathrm{em}} = X^{\mathrm{in}} W^{\mathrm{em}}, X^{(1)} = X^{\mathrm{em}} + X^{\mathrm{pos}}, \tag{3}$$

where $X^{\mathrm{pos}}$ is a trainable positional vector. For the $l$-th layer, the $Q, K, V$ are defined as:

$$Q^{(l)} = X^{(l)} W^{Q(l)}, \quad K^{(l)} = X^{(l)} W^{K(l)}, \quad V^{(l)} = X^{(l)} W^{V(l)}. \tag{4}$$

The attention matrix $\mathrm{Attn}^{(l)}$ and its subsequent output $X^{\mathrm{qkv}(l)}$ for the $l$-th layer is computed as:

$$\mathrm{Attn}^{(l)} = \mathrm{softmax}\left(\frac{Q^{(l)} K^{(l)T}}{\sqrt{d_k}}\right) \text{ (with mask)}, \quad X^{\mathrm{qkv}(l)} = \mathrm{Attn}^{(l)} V^{(l)}. \tag{5}$$

The output of the $l$-th attention layer is obtained as:

$$X^{\mathrm{ao}(l)} = \mathrm{LN}(X^{(l)} + X^{\mathrm{qkv}(l)}W^{\mathrm{attn},l}), \quad X^{(l+1)} := X^{\mathrm{do}(l)} = \mathrm{LN}(\mathrm{FNN}(X^{\mathrm{ao}(l)}) + X^{\mathrm{ao}(l)}), \quad (6)$$

where "LN" refers to Layer Normalization and "FN" represents a Fully Connected Network. The final output is obtained by projecting the output of the last layer $X^{\mathrm{do}(L)}$ using a linear projection layer, followed by a softmax operation and argmax to obtain the predicted token.

For the basic experimental setups, we use cross-entropy loss on the last token of the sequence and optimize the model using Adam with weight decay. The specific hyperparameters and training details are provided in Appendix A.

## 4  Two Phases of Solutions for Composite Functions

In this section, we investigate the mechanisms of how a transformer learns the compositional tasks, especially on the OOD tasks.

**Experimental Setup.** For the 16 anchor pairs, the anchor pair (4, 3) is held out as an unseen pair during the training, while pair (3, 4) is assigned as non-inferential mapping, i.e., we set the designated target mapping $\mathcal{M}_{(3,4)}(x) = x - 6$ (the inferential mapping is $x - 10$), seen in the training set. The other 14 anchor pairs are set as inferential mappings seen in the training set. Three possible solutions for pair (4, 3) may be learned after enough training: i) inferential solution based on the 14 anchor pairs with inferential mapping, ii) symmetric solution based on the anchor pair (3, 4), iii) other non-inferential solutions. See illustration in Fig. 1(b).

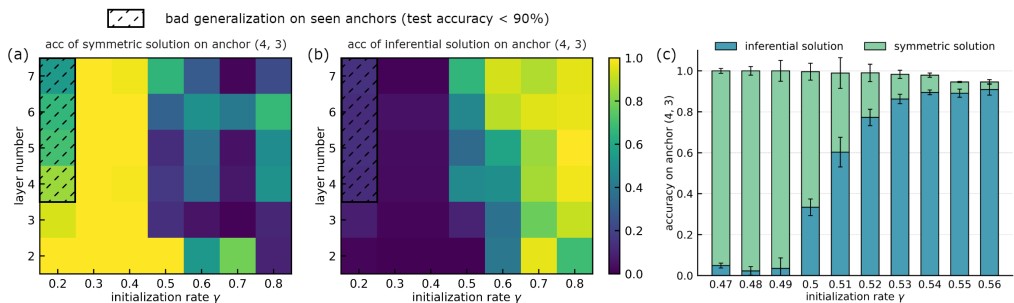

Figure 2: (a,b) Phase diagram of generalization performance on the *unseen* anchor (4, 3). (a) The model's test accuracy based on the symmetric mapping. (b) The model's test accuracy based on the inferential mapping. The abscissa represents the initialization rate $\gamma$, which corresponds to the standard deviation $(1/d_{\mathrm{in}})^{\gamma}$ of a normal distribution with a mean of 0 used for parameter initialization. The ordinate represents the depth of the transformer model. The shadow zones indicate the test accuracy on seen anchors is less than 90%. (c) Comparison of accuracy on the unseen anchor (4, 3) for both the inferential and symmetric solutions across different initialization rates $\gamma$ on GPT-2. The error bars represent the standard deviation across 4-time runs.

An important and interesting question is: *which solution is learned for unseen pair (4, 3)?* Experiments show that different initialization scales lead to different solutions for the unseen pair (4, 3) with different model depths. Fig. 2(a) shows the phase diagram assuming the symmetric mapping ($\mathcal{M}_{(4,3)}(x) = x - 6$) as ground truth, with initialization scale on the abscissa and model depth on the ordinate. A large initialization scale (small $\gamma$) leads to the symmetric solution, indicated by yellow zones with nearly 100% accuracy. Fig. 2(b) assumes the inferential mapping ($\mathcal{M}_{(4,3)}(x) = x - 10$) as ground truth, showing that a small initialization scale (large $\gamma$) favors the inferential solution. Larger initialization scales with deep models result in poor generalization even on seen anchors (shadow zone). This pattern was consistent across trials: for each random seed, models with different initializations and depths were trained using 9 learning rates (uniformly sampled in $[1 \times 10^{-4}, 3 \times 10^{-4}]$), and the highest accuracy for each of the two mappings was selected from these rates. The highest accuracies were then averaged over 3 random seeds. We conclude that small initializations favor the inferential solution, while large (but not very large) initializations favor the symmetric solution.

To validate the generality of these findings, we extend our experiments to the multi-head GPT-2 model. As shown in Fig. 2(c), the model exhibits similar generalization performance trends on the unseen pair (4, 3) across different initialization rates $\gamma$. The range of $\gamma$ variation is smaller in this setup, likely due to the difference in model parameter scale between the GPT-2 and the transformer models used in the previous experiments.

We also find that increasing the learning rate and weight decay coefficient within a certain range is helpful for the transformer to learn the inferential solution on the unseen anchor. Please refer to Appendix E for experimental phenomena and detailed discussion.

## 5 Mechanisms of Models in Two Phases

In this section, we analyze mechanisms underlying different learning solutions including: i) information flow via attention matrix; ii) information representation in vector space; iii) model complexity from different perspectives. The key message is as follows. A model with small initialization needs to gradually increase its capacity by enlarging the parameter scale, therefore, to use the least optimization cost to fit the training data, the model only needs to learn four single anchor functions to obtain inferential mappings. As the initialization scale increases, the model tends to learn ten mappings, treating symmetric anchor pairs as equivalent, and memorizes the mapping between each anchor-key item pair and its corresponding output. When the initialization becomes even larger, the model forgoes learning general patterns and instead merely memorizes the output associated with each individual data point. This causes the model to lose generalization ability, even on the anchors it has encountered during training.

### 5.1 Information Transmission and Fusion Mechanisms

In this section, we study the information transmission and fusion mechanisms occurring in the attention matrix through information flow analysis. As shown in Fig. 3 (a,c), we present the information flow of two-layer networks corresponding to the two solutions. We use the same input sequence to test the output with the key item 99 and the unseen anchor pair $(4, 3)$. The thickness of the line connecting the $j$-th token in Layer $l$ and the $k$-th token in Layer $l+1$ represents the value of the attention matrix $\text{Attn}^{(l)}$ at position $(k, j)$. For ease of observation, we manually annotate the information flow that significantly contributes to the output of the last token using different colors.

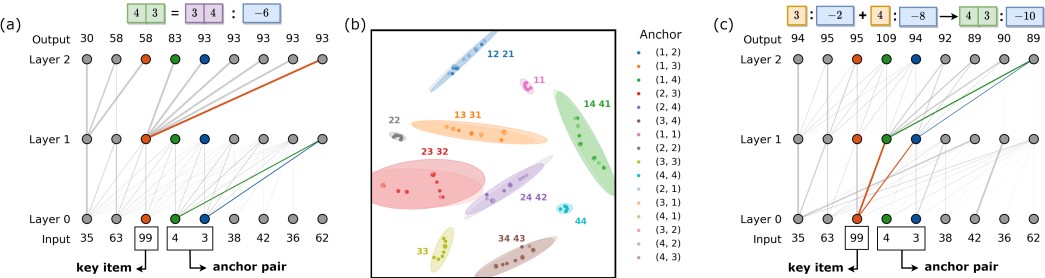

Figure 3: (a, c) Information flow in the two-layer networks of symmetric and inferential solutions. The input sequence shown in the figure represents the test sample, with key items and anchor positions annotated. For each layer's attention matrix, we illustrate the mechanisms of information transmission and fusion through the information flow. The thickness of the line represents the corresponding value in the attention matrix $\text{Attn}^{(l)}$. We use different colors to mark the key item and the two single anchors, and highlight the attention connections that significantly contribute to the final output. The final output sequence represents the model's output. (a) Symmetric solution. (c) Inferential solution. (b) T-SNE visualization of vectors $X^{\text{ao}(1)}$ of 10,000 input sequences with different anchor-key item pairs. Symmetric anchor pairs have similar colors in different shades.

For symmetric solutions, we find that the model's attention mechanism plays a key role in enabling the learning of consistent mappings for symmetric anchor pairs. In the first layer of the network, the attention mechanism combines the information of the two anchors and moves it to the last token of the sequence. This allows the model to learn identical embeddings for symmetric anchor pairs,

such as (3, 4) and (4, 3), as illustrated in Fig. 3(a). In the second layer, the information of the key item is broadcast to the last token and combined with the anchor information obtained from the first layer by the residual connection. This enables the model to produce the final output based on the combined information from the anchors and the key item.

To verify the symmetric nature of the learned representations after the first-layer attention, we visualize the vectors $X^{\mathrm{ao}(1)}$ of 10,000 input sequences with different anchor-key item pairs using t-SNE. Fig. 3(b) shows that symmetric anchor pairs are clustered together in the $X^{\mathrm{ao}(1)}$ vector space, confirming that the model learns to map them to similar representations.

For inferential solutions, the information transmission mechanism is different. As shown in Fig. 3(c), in the first layer, the key item information is moved to the positions of the two single anchors, and each anchor is combined with the key item information separately. This allows the model to learn individual mappings for each single anchor. In the second layer, the anchor tokens (now containing the combined information from the anchors and the key item) are moved to the last token and combined to produce the final output. This combination mechanism enables the model to learn inferential solutions on the composite anchors.

## 5.2 Divergence in Fused Vector Representations across Two Phases

To further investigate the mechanistic differences between symmetric and inferential solutions, we examine the divergence in vector representations after the fusion of anchor pair and key item information in both types of solutions. Specifically, we examine the cosine similarity between the output vectors of the second attention layer's last token (i.e., the last token of $X^{\mathrm{ao}(2)}$) for different anchor-key item combinations. We denote these output vectors as $\boldsymbol{v}(x; a_1, a_2)$, where $x$ is the key item and $(a_1, a_2)$ is the anchor pair of the input sequence[2].

As illustrated in Fig. 4(a, b), the heatmaps display the cosine similarity between output vectors for various anchor-key item pairs. Both the abscissa and ordinate represent key item values ranging from 30 to 40. The lower and upper triangles of the heatmap correspond to the cosine similarity matrices between $\boldsymbol{v}(x_1; 3, 3)$ and $\boldsymbol{v}(x_2; 2, 2)$, and between $\boldsymbol{v}(x_1; 1, 2)$ and $\boldsymbol{v}(x_2; 1, 3)$, respectively, where $x_1$ and $x_2$ are the corresponding key item values on the axes. The red boxes highlight positions where the inferential targets are the same, for example, the red highlight in the first column in Fig. 4(a) indicates $f(36; 3, 3) = f(30; 2, 2)$, where $f(\cdot; \cdot, \cdot)$ is defined in Equation (1).

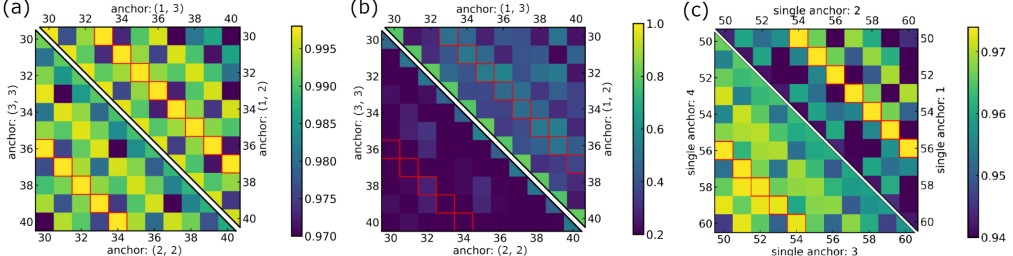

Figure 4: Cosine similarity heatmaps for vector representations in different solutions. Each axis represents a selected anchor pair (labeled on the axis), with the value on the coordinate axis representing the value of the key item.The color indicates the cosine similarity between specific vectors defined in each subplot. Red boxes highlight positions where the target outputs obtained by the anchors on the abscissa and ordinate are the same for the corresponding key items. (a, b) Cosine similarity between the output vectors of the second attention layer's last token (the last token of $X^{\mathrm{ao}(2)}$) for different anchor-key item pairs in (a) inferential and (b) symmetric solutions. (c) Cosine similarity between the rows of the second layer Value matrix ($V^{(2)}$) corresponding to the first anchor's position across different anchor-key item pairs for inferential solutions.

For inferential solutions, if two anchor-key pairs $(x_1; a_1, a_2)$ and $(x_2; b_1, b_2)$ have the same output, i.e., $f(x_1; a_1, a_2) = f(x_2; b_1, b_2)$, then their fused vectors $\boldsymbol{v}(x_1; a_1, a_2)$ and $\boldsymbol{v}(x_2; b_1, b_2)$ are nearly

---

[2]For simplicity, we assume that the position of the anchor-key item combination and other items do not significantly affect the output vector.

parallel. In contrast, for symmetric solutions, the pairwise similarity between vectors $\boldsymbol{v}(x; a_1, a_2)$ is always low even for those with the same output. This suggests that the last FNN layer of the symmetric solution memorizes all possible projections from $X^{\mathrm{ao}(2)}$ to the decoder layer output. Conversely, the last FNN of the inferential solution only needs to learn fewer projections, as the same output is represented by similar vectors in the $X^{\mathrm{ao}(2)}$ space.

Learning inferential solutions relies on the model's ability to infer the operation of each individual anchor. To verify the alignment between the single anchor operations in the inferential mechanism and the defined single-anchor function $g(\cdot; \cdot)$ in Equation (2), we investigate the vector representations after the fusion of the first single anchor and the key item. We focus on the vector representation of the second layer Value matrix $V^{(2)}$ at the position of the first single anchor, denoted as $\boldsymbol{u}(x; a_1)$, with key item $x$ and first anchor $a_1$. Similar to Fig. 4(a, b), the red boxes highlight the positions where the targets obtained by the single anchors $a_1$ and $a_2$ on the abscissa and ordinate are the same for the corresponding key items $x_1$ and $x_2$, i.e., $g(x_1; a_1) = g(x_2; a_2)$. As shown in Fig. 4(c), the information fusion of single anchors and key items and the explicitly defined single anchor operations align well. This provides stronger evidence for the model's ability to learn the mapping of each single anchor.

## 5.3 Model Complexity: A Key Factor in Phase Transitions

Large initialization scales endow models with high complexity, allowing them to fit training data with minor parameter changes, as seen in FNNs in the linear regime (e.g., Neural tangent kernel) (Jacot et al., 2018; Luo et al., 2021). Conversely, models with small initialization scales start with low complexity and gradually increase it during training. In small initialization FNNs, the input weights of different neurons often cluster along a few isolated orientations, which is referred to as condensations (Zhou et al., 2021; Zhang et al., 2021, 2022). This phenomenon of parameter condensation is closely related to the model's complexity and its ability to learn the underlying structure of the data. To better understand the mechanisms behind inferential and symmetric solutions, we investigate the degree of parameter condensation.

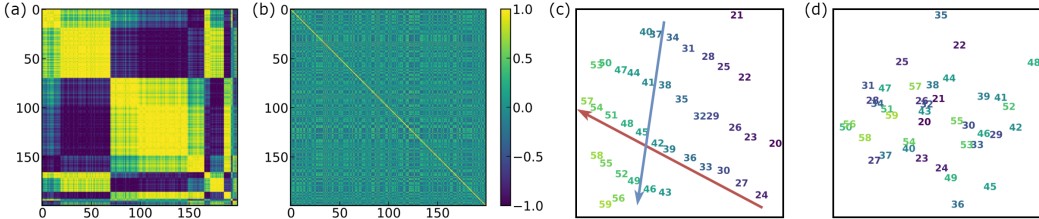

Figure 5: (a, b) Cosine similarity of neurons in the $W^{Q(1)}$ matrix. The abscissa and ordinate both represent neuron index. (a) Inferential solution with small initialization. (b) Symmetric solution with large initialization. (c, d) Visualization of the embedding space using t-SNE for different initialization scales. (c) Inferential solution with small initialization. The embedded tokens seem to form arithmetic sequences with common differences of 3 (red arrow) and 4 (blue arrow) along the two directions. (d) Symmetric solution with large initialization. Please refer to Appendix D for more detailed experimental results under different model depths and initialization rates $\gamma$.

**Condensation.** We examine the condensation of matrix $W^{Q(1)}$. The similarity between the $i$-th and the $j$-th neurons input weight is calculated by cosine similarity, i.e., $\frac{W^{Q(1)}[i,:] \cdot W^{Q(1)}[j,:]}{||W^{Q(1)}[i,:]||_2 ||W^{Q(1)}[j,:]||_2}$. As shown in Fig. 5, for the inferential solution (Fig. 5(a)), the neuron weights condense in a few directions, suggesting low complexity within the model. In contrast, for the symmetric solution (Fig. 5(b)), there is no obvious condensation of neuron parameters, indicating high complexity within the model.

However, even with small initialization, not all parameters exhibit significant condensation. For the word embedding matrix $W^{\mathrm{em}}$, to distinguish the meanings of different tokens, different neuron weights have different directions, i.e., neurons do not condense. Nevertheless, the parameter matrix still exhibits a clear tendency towards low complexity.

**Structural Features of the Word Embedding Matrix.** To gain further insights into the learning mechanisms of inferential solutions, we analyze the structure of the model's embedding matrix. We visualize the embedding space using t-SNE. As shown in Fig. 5(c,d), for small initialization scales (Fig. 5(c)), we observe a clear ordinal structure in the embedding space, with the embeddings of the input tokens arranged according to their numerical value. This also suggests a low-complexity tendency in the word embedding matrix, requiring the model to capture the relative ordinal relationships between different tokens. However, this ordinal structure is not present for large initialization scales (Fig. 5(d)).

It is worth noting that the relative ordinal relationship in the inferential solution is not a simple numerical magnitude relationship of the corresponding tokens. We observe that this ordinal relationship may originate from the definition of the four single anchors, where the differences between the operations of any two single anchors can be obtained by the addition of the basic elements 3 and 4. This arrangement is consistent with the numbers being ordered with intervals of 3 (red arrow) and 4 (blue arrow) from two directions in the embedding space. To further verify the low complexity bias of the word embedding matrix, we visualize the eigenvalues of the covariance matrix of the embedding vectors and its evolution process, the results are shown in Appendix D.3.

## 6 Further Verification on Realistic Tasks

We validated the performance of models with different initialization scales and weight decay settings across a series of compositional and reasoning tasks. Below, we introduce each task and the corresponding results. Please refer to the Appendix F for a detailed introduction of each dataset.

**Compositional tasks: SCAN and COGS.** For the SCAN dataset (Lake and Baroni, 2018), we selected the "Generalizing composition across primitive commands" task, where the "turn left" command only appears in single-command mappings and is trained alongside other composite commands. We assess the model's generalization ability on composite commands that include the "turn left" command. For the COGS dataset (Kim and Linzen, 2020), we evaluate the model's in-distribution and out-of-distribution generalization performance after training on the same training set.

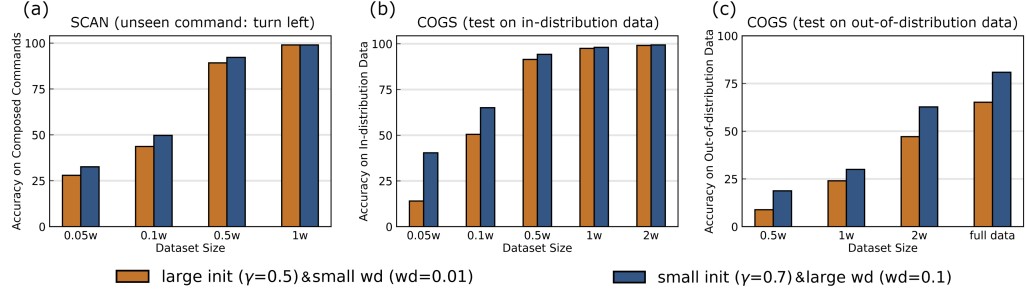

Figure 6: Performance comparison of models with different initialization scales and weight decay coefficients on compositional tasks. (a) For the SCAN task, we assess the generalization ability on composite commands that include the "turn left" command. (b, c) For the COGS task, we evaluate (b) in-distribution and (c) out-of-distribution generalization after training on the same dataset. Small initialization and large weight decay (blue) consistently outperform large initialization and small weight decay (orange) across different tasks and data scales. The parameters are initialized following a zero-mean normal distribution with a standard deviation of $d_{in}^{-\gamma}$.

As shown in Fig. 6, we display the generalization performance of models with different initialization scales and weight decay coefficients across various data sizes. Small initialization and large weight decay consistently outperform large initialization and small weight decay across different task types and data scales. Notably, in the COGS task, even when the in-distribution generalization of both settings (with 20k training data) reaches over 99%, the difference in out-of-distribution generalization remains significant.

**Reasoning tasks: PrOntoQA.** PrOntoQA (Saparov and He, 2023) is a synthetic multi-step reasoning dataset where each data point assigns hierarchical relationships among objects and requires the

model to determine whether a multi-step reasoning chain is correct. Fig. 7 illustrates the convergence rates and generalization errors[3] with respect to data scale for models with large initialization (and small weight decay, Fig. 7(a)) and small initialization (and large weight decay, Fig. 7(b)). An interesting phenomenon is observed for models with large initialization (small weight decay): as the data size increases, the convergence rate first decreases and then increases. When the data size is small, the model tends to fit the data through memorization. Therefore, as the data size increases, the training difficulty increases (i.e., the training speed slows down), and the model's generalization ability is poor. As the data size grows further, the model, constrained by its complexity, can no longer memorize all the data and thus shifts to fitting the data through reasoning. This leads to an increase in fitting speed and results in better generalization. In contrast, models with small

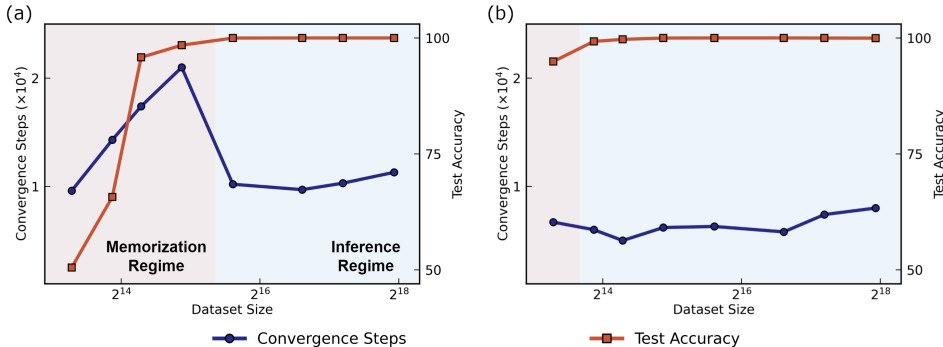

Figure 7: Performance comparison of models with different initialization scales and weight decay coefficients on PrOntoQA. (a) Convergence steps and test accuracy for large initialization ($\gamma = 0.5$) and small weight decay (WD = 0.01). (b) Convergence steps and test accuracy for small initialization ($\gamma = 0.7$) and large weight decay (WD = 0.1). Models with large initialization initially struggle with memorization before improving as data size increases, whereas small initialization models maintain faster convergence and better generalization. The parameters are initialized following a zero-mean normal distribution with a standard deviation of $d_{in}^{-\gamma}$.

initialization (large weight decay) inherently prefer to fit the data through reasoning, leading to faster convergence and better generalization at the same data scale compared to models with large initialization (small weight decay).

# 7 Discussion

**Conclusion.** In this work, we investigate the influence of parameter initialization scale on the solutions learned by transformers for compositional tasks. We discover a phase diagram of model solutions, where small initialization scales lead to inferential solutions that capture the underlying compositional primitives, while larger initialization scales result in symmetric solutions that memorize mappings. To explain these findings, we analyze the information flow and vector representations within the model, revealing distinct mechanisms for inferential and symmetric solutions. Inferential solutions exhibit low complexity and learn individual mappings for single anchors, while symmetric solutions directly combine anchor information without learning the compositional structure. We further extend our experiments to GPT-2, and verify that the insights remain valid for more complex architectures on various real-world datasets.

**Limitation and Future Work.** The key limitation of our work is that the experiments and analyses are based on synthetic data, which may not fully capture the complexities of real-world datasets and tasks. Although some of our conclusions have also been validated in models like GPT-2, further verification on a wider range of models is necessary. Going forward, we plan to extend our investigation to real-world datasets and tasks, to bridge the gap between our theoretical understanding and practical application. This could involve leveraging Mixture of Experts (MoEs) to design networks with different initialization scales for different expert models.

---

[3]During testing, we only evaluate the model's accuracy in judging hierarchical relationships. Thus, the model's random guessing accuracy is 50%.

## Acknowledgments

This work is sponsored by the National Key R&D Program of China Grant No. 2022YFA1008200, the National Natural Science Foundation of China Grant No. 92270001(Z. X.), 12371511 (Z. X.), 12422119 (Z. X.), 12101402 (Y. Z.), the Lingang Laboratory Grant No. LG-QS-202202-08 (Y. Z.), Shanghai Municipal of Science and Technology Major Project No. 2021SHZDZX0102 (Z. X., Y. Z.), and the HPC of School of Mathematical Sciences and the Student Innovation Center, and the Siyuan-1 cluster supported by the Center for High Performance Computing at Shanghai Jiao Tong University, Key Laboratory of Marine Intelligent Equipment and System, Ministry of Education, P.R. China. This work was partially supported by SJTU Kunpeng&Ascend Center of Excellence.

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

# A Experimental Setups

For Fig. 2(a,b), Fig. 3, Fig. 4, Fig. 5, Fig. 10, Fig. 11, Fig. 12 and Fig. 13, we train the transformer model on a dataset of 900,000 samples, with each input sequence having a length of 9 tokens. The key items in the input sequences are randomly sampled from the range 20-99. The model architecture consists of a varying number of layers, a single attention head, an embedding dimension of 400, a feedforward dimension of 1200, and key and value dimensions of 200 each. The model parameters are initialized using a normal distribution with a mean of 0 and a standard deviation of $(1/d_{\text{in}})^\gamma$, where $d_{\text{in}}$ is the input dimension of the parameter and $\gamma$ is the initialization rate. We employ the AdamW optimizer with a learning rate of 1e-5, epsilon of 1e-8, weight decay of 0.01, and $\beta_1$ and $\beta_2$ values of 0.9 and 0.999, respectively. The model is trained for 210 epochs using a batch size of 2048 and a gradient clipping maximum norm of 1. The learning rate is scheduled using a warm-up period followed by cosine decay, with a warmup period of 10 epochs, a multiplier of 25 (if there is no special instruction), a cosine decay number of epochs of 200, and a minimum learning rate of 1e-5. For Fig. 3(a, b), Fig. 4(b), Fig. 5(b, d), we use a two-layer transformer with initialization rate $\gamma = 0.5$. For Fig. 3(c), Fig. 4(a, c), Fig. 5(a, c), we use a two-layer transformer with initialization rate $\gamma = 0.8$.

For Fig. 2(c), Fig. 6, Fig. 7, Fig. 17 we employ the architecture of GPT-2 in our experiments. For Fig. 2(c), to ensure consistency between the model structure and the settings described in the paper, we adopt the post-norm configuration. We have also conducted experiments using the pre-norm setting and obtained consistent conclusions. Apart from the differences in the model architecture, we retain the same data, training strategy, parameter initialization, and other settings as mentioned in the previous sections.

For Fig. 14, Fig. 15, Fig. 16, we train the networks using initialization rate $\gamma = 0.5$ with different learning rates and weight decay coefficients. The learning rate is scheduled using a warm-up period followed by cosine decay, with a warmup period of 10 epochs, a multiplier chosen from the set {5, 10, 20, 40, 60}, a cosine decay number of epochs of 200, and a minimum learning rate of 1e-5.

**Experiments Compute Resources.** The experiments were conducted on a server with the following configuration:

- 48 AMD EPYC 7352 24-Core Processors, each with 512KB of cache
- 251GB of total system memory
- 8 NVIDIA GeForce RTX 4090 GPUs with 24GB of video memory each
- The experiments were run using Ubuntu 22.04 LTS operating system

# B  Illustrative Examples of Key Definitions

## B.1  Examples of Two-anchor composite function

In this paper, we use an anchor set $A = \{1, 2, 3, 4\}$ containing four elements. Each anchor corresponds to a specific function:

$$g(x; 1) = x + 5, \quad g(x; 2) = x + 1, \quad g(x; 3) = x - 2, \quad g(x; 4) = x - 8.$$

Suppose we have an input sequence $X = (23, 1, 2, 43, 46, 74, 54, 44, 72)$. In this sequence, the 2nd and 3rd items (i.e., items 1 and 2) belong to the anchor set $A$, thus forming an anchor pair $(1, 2)$. The item 23, immediately preceding the anchor pair, is called the key item.

For this input sequence $X$, the computation process of the two-anchor composite function is as follows:

$$f(X) = f(23; 1, 2) = g(g(23; 1); 2) = g(23 + 5; 2) = 28 + 1 = 29.$$

As we can see, the two-anchor composite function first passes the key item 23 into the function $g(x; 1) = x + 5$ corresponding to the first anchor 1, obtaining the result 28; then it passes 28 into the function $g(x; 2) = x + 1$ corresponding to the second anchor 2, finally obtaining the output 29.

## B.2  Examples of Data Generation

In the experiments of this paper, we use four anchors (i.e., 1,2,3,4) and numbers sampled from 20 to 99 to construct the input dataset. Each input sequence includes an anchor pair, a key item (immediately preceding the anchor pair), and some noise items unrelated to the target.

For example, we can construct an input sequence as follows:

$$X = (52, 33, 36, 2, 4, 78, 92, 24, 58).$$

In this example, the anchor pair is $(2, 4)$, the key item is 36, and 52,33,78,92,24,58 are noise items.

For a given input sequence $X$, we stipulate that its target is only related to the key item and the anchor pair. Furthermore, we artificially assign mappings to the anchor pairs and take the output value of the key item under the mapping corresponding to the anchor pair as the target of this sequence. In the default case, we use the composite anchor function corresponding to the anchor pair as the mapping corresponding to this anchor pair, and this mapping is called the inferential mapping (derived by composing single anchor mappings). In some special cases, we may modify the mapping corresponding to the anchor pair so that it does not satisfy the corresponding composite anchor function, and such mappings are called non-inferential solutions. The detailed examples are shown in Section B.3.

## B.3  Examples of Mapping Type of an anchor pair

For an anchor pair $(a_1, a_2)$, its mapping $\mathcal{M}_{(a_1, a_2)}(\cdot)$ can have the following types:

**Inferential mapping.** If the designated target mapping of the anchor pair $(a_1, a_2)$ is consistent with the output of the two-anchor composite function, i.e., $\mathcal{M}_{(a_1, a_2)}(x) = f(x; a_1, a_2)$, then it is called an inferential mapping.

For example, for the anchor pair $(1, 2)$ and key item $x$, if the target output is $f(x; 1, 2) = g(g(x; 1); 2) = (x + 5) + 1 = x + 6$, then the mapping of $(1, 2)$ is an inferential mapping.

**Non-inferential mapping.** If the designated target mapping of the anchor pair $(a_1, a_2)$ is inconsistent with the output of the two-anchor composite function, i.e., $\mathcal{M}_{(a_1, a_2)}(x) \neq f(x; a_1, a_2)$, then it is called a non-inferential mapping.

For example, for the anchor pair $(1, 2)$ and key item $x$, if the target output is designated as $x + 10$, rather than $f(x; 1, 2) = g(g(x; 1); 2) = (x + 5) + 1 = x + 6$, then the mapping of $(1, 2)$ is a non-inferential mapping.

**Symmetric mapping.** If the designated target mapping of the anchor pair $(a_1, a_2)$ is consistent with the mapping of its symmetric anchor pair $(a_2, a_1)$, i.e., $\mathcal{M}_{(a_1, a_2)}(x) = \mathcal{M}_{(a_2, a_1)}(x)$, then it is called a symmetric mapping.

For example, suppose the mapping of the anchor pair $(3, 4)$ is designated as $\mathcal{M}_{(3,4)}(x) = x + 10$. If the mapping of the anchor pair $(4, 3)$ is also designated as $\mathcal{M}_{(4,3)}(x) = x + 10$, then the mapping of $(4, 3)$ is a symmetric mapping, because it is exactly the same as the mapping of $(3, 4)$.

For a given model, we define its inferential (non-inferential, symmetric) accuracy on a specific anchor pair as the accuracy obtained when using the inferential (non-inferential, symmetric) mapping of that anchor pair as the target. In particular, if a model exhibits a tendency to generate outputs that align with the inferential (non-inferential, symmetric) mapping for an anchor pair, we adopt the convention of stating that the model has learned the inferential (non-inferential, symmetric) solution for that specific anchor pair.

## B.4    Examples of Generalization

As introduced in the main text, there are two types of generalization in this paper: generalization on data and generalization on task. Here we provide some examples to illustrate their differences, also shown in Fig. 8.

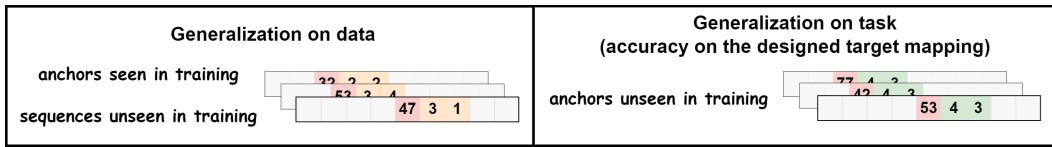

Figure 8: Illustration of the two types of generalization studied in this paper. Generalization on data tests a model's performance on sequences with seen anchors but unseen combinations of key items and anchor pairs. Generalization on task evaluates a model's ability to infer the mappings for unseen anchor pairs.

**Generalization on data.** The test set for evaluating generalization on data consists of sequences with anchor pairs seen in the training set, but the sequences (detailed splitting method is shown in Appendix C) are not seen in training. Models with good generalization on data should be able to correctly predict the targets for these test sequences.

**Generalization on task.** To test generalization on task, we need to construct sequences with anchor pairs not seen (anchor pair (4, 3)) during training and designate their target mappings. If the model can output targets consistent with the designated mappings for (4,3), then it achieves good generalization on task. It is worth noting that the unseen anchor pair (4, 3) does not have a natural target. Therefore, we can calculate its accuracy under different designated target mappings to reflect the model's preference for the learned solutions under various settings.

## C   Dataset Splitting Method

A straightforward division based on data ranges proves to be impractical. To illustrate, consider a scenario where the range of key items in the training set is denoted as $[a, b]$, while in the test dataset, it is represented as $[b+1, c]$. The encoding of data within the interval $[b+1, c]$ is not learned during the neural network training process. As a result, the neural network fails to produce the key item output for the test dataset.

To address this issue, we divide the dataset based on the value and the position of the key item, as shown in Fig. 9. Consider a task with an input sequence of length $n$. For an input sequence in the training dataset, an item $x$ can be placed in the $pos$-th position of such input sequence only when $\mathrm{mod}(x, n-2) \neq pos$. For an input sequence in the test dataset, if the token at the $i$-th position is a key item, then an item $x$ can be placed in the $pos$-th position of such input sequence only when $\mathrm{mod}(x, n-2) = pos$. It is important to note that the test data and training data are not completely separated in terms of values. However, when the positions of the key items are the same, the corresponding test data and training data do not overlap.

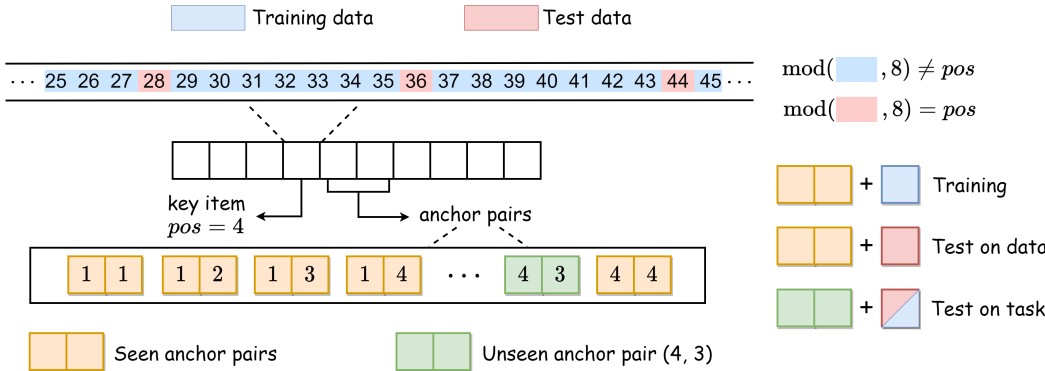

Figure 9: Illustration of the dataset splitting method based on the value and position of the key item. The training data and test data are divided according to the modulo operation on the key item value and its position in the input sequence.

We further define two types of generalization based on this dataset splitting method. For training data, we use pairs of seen anchors and training data (training key items) for training. Regarding data generalization, we test the model using pairs of seen anchors and test data (test key items) to evaluate the model's ability to generalize to different items within seen composite mappings. For task generalization, we use pairs of unseen anchors with test data or training data to evaluate the model's performance on masked composite mappings. It is important to note that for task generalization, we can test the accuracy of the anchor pairs with different ground truth mappings. This accuracy reflects the model's preferred mappings for these anchor pairs.

# D  Detailed Results for Model Complexity with Different Initialization

## D.1  Detailed Results for Condensation

In order to show the parameter condensation at different depths and different initialization scales, we plotted the cosine similarity of the $W^{Q(1)}$ matrix neurons corresponding to the phase diagram in Fig. 2. As shown in Fig. 10, for each subgraph, we group the weights of two neurons with cosine similarity greater than 0.7 into the same category (adjacent index in the heat map) to highlight the condensation properties of neurons. It is easy to see that as the initialization rate increases (the initialization scale becomes smaller), the $W^{Q(1)}$ matrix neurons show an obvious condensation trend, implying that the model complexity decreases.

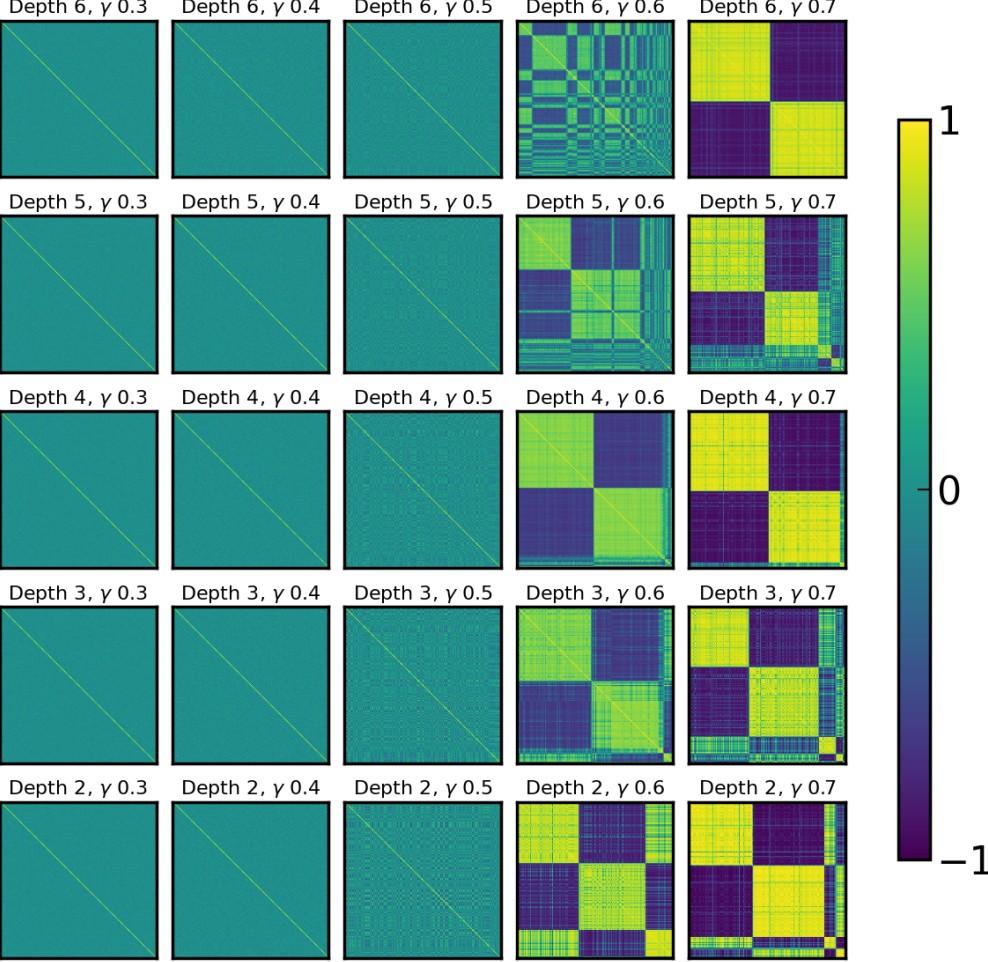

Figure 10: Cosine similarity matrices of $W^{Q(1)}$ neurons at different depths and initialization scales. Each subgraph represents a different depth (from 2 to 6, bottom to top) and initialization rate $\gamma$ (from 0.3 to 0.7, left to right). Colors indicate the cosine similarity between neurons, with warmer colors representing higher similarity. The neurons are grouped by cosine similarity greater than 0.7 to highlight the condensation properties. As the initialization scale increases, neurons exhibit more condensation, indicating decreased model complexity.

## D.2 Detailed Results for the Structural Features of the Word Embedding Matrix

Similar to the previous subsection on condensation, we examine the structural changes in the model's word embedding matrix as the initialization scale and model depth vary. We visualize the parameter matrix using t-SNE. As shown in Fig. 11, as the initialization rate $\gamma$ increases, the word embedding matrix gradually exhibits distinct structural features. These structural features enable the model to accurately represent the numerical meaning of different items. Additionally, a high degree of structure indicates low complexity in the word embedding matrix (i.e., low matrix rank), aligning with the low complexity hypothesis for models with small initialization scales.

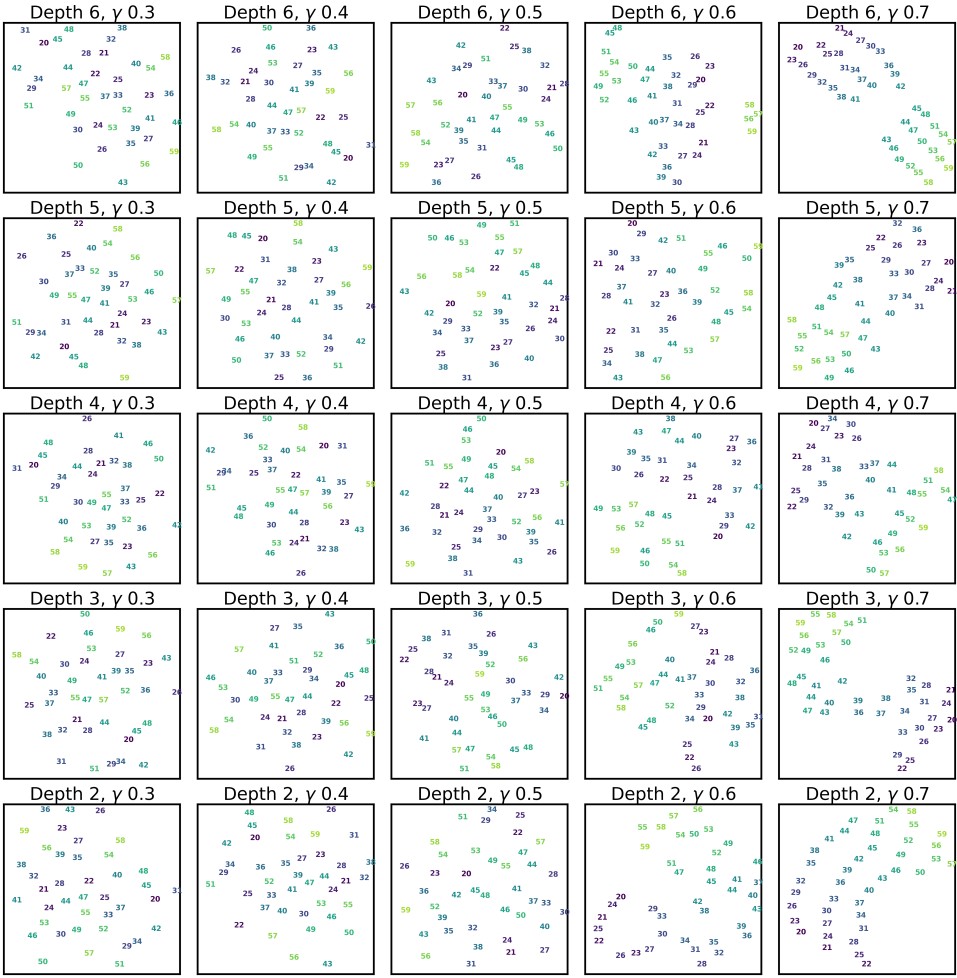

Figure 11: Visualization of the embedding space using t-SNE at different depths and initialization scales. Each subgraph represents a different depth (from 2 to 6, bottom to top) and initialization rate $\gamma$ (from 0.3 to 0.7, left to right). As the initialization scale increases, the word embedding matrix gradually exhibits distinct structural features, indicating decreased model complexity.

## D.3 Detailed Rank Analysis for Different Weight Matrices

To further verify the low complexity bias of the word embedding matrix, we visualize the eigenvalues of the covariance matrix of the embedding vectors (Fig. 12(a)). For small initialization scales, the covariance matrix has a small number of large eigenvalues, indicating that the model learns a low-dimensional representation of the input tokens that captures their ordinal relationships. This low-dimensional representation facilitates the learning of inferential solutions by aligning with the

underlying structure of the single anchor operations. In contrast, for large initialization scales, the eigenvalues are more evenly distributed, suggesting that the model learns a more distributed representation that does not effectively capture the ordinal structure. This lack of structure in the embedding space hinders the model's ability to learn the relationships between the input tokens and the underlying single anchor operations, leading to the learning of symmetric solutions.

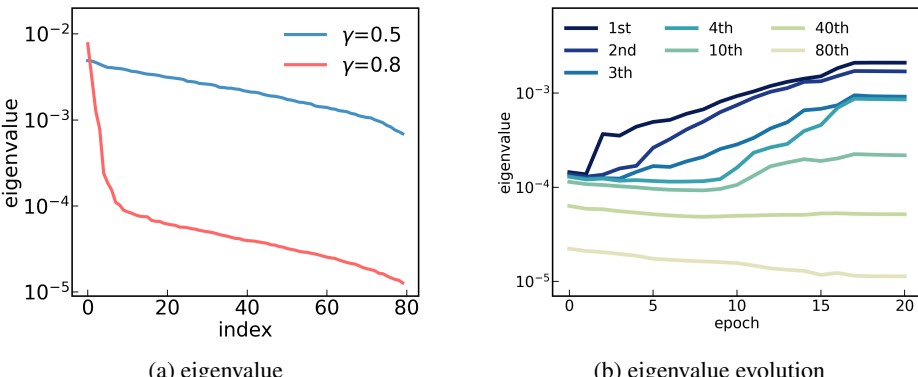

|(a) eigenvalue|(b) eigenvalue evolution|

Figure 12: Eigenvalues of the covariance matrix of the embedding matrix for different initialization scales and the evolution process of the eigenvalues of the small initialization model. Left: Eigenvalues of the covariance matrix of the embedding vectors for different initialization scales. The abscissa is the eigenvalue index, and the ordinate is the eigenvalue. Colors represent different initialization scales. The definition of the initial scale $\gamma$ is consistent with Fig. 2. Right: The evolution process of the eigenvalues of specific indexes of the small parameter initialization model as the training progresses.

Meanwhile, we investigated the evolution of the eigenvalues corresponding to specific indices of the small initialization model during training. As shown in Fig. 12(b), the model gradually increases its complexity over the training process. Specifically, the model first increases the value of the largest eigenvalue, while the other eigenvalues remain almost unchanged in the initial phase of training. As training progresses, the model requires more eigen-directions to fit the data, leading to a subsequent increase in the other eigenvalues. Once the model complexity increases sufficiently to fit the training data, it ceases to further increase its complexity. This gradual increase in complexity ensures that the model maintains the lowest possible complexity necessary to fit the data well, enabling it to learn fundamental operations rather than merely memorizing the training data.

To further validate the low complexity of the model under small initialization, we present the singular value distributions of the weight matrices across various linear layers for both models with different initializations, as shown in Fig. 13. It is evident that, for the small initialization model, the first few singular values are significantly larger than the remaining ones. In contrast, this distinct difference is not observed in the model with large initialization. This indicates the pervasive low-complexity characteristic of the internal parameters in the small initialization model.

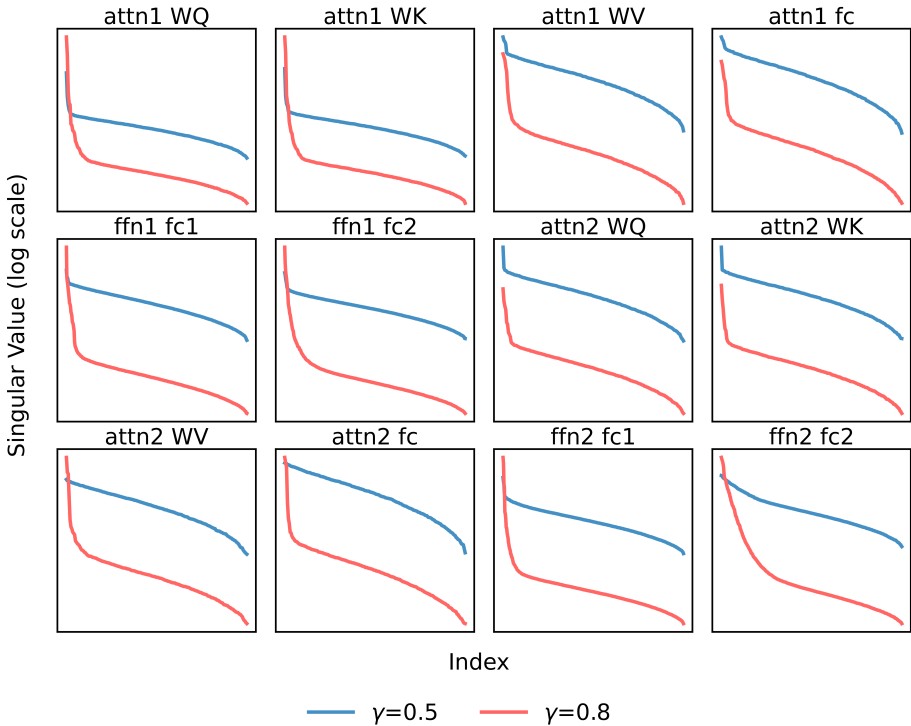

Figure 13: Singular value distributions of the weight matrices across various linear layers for models with different initializations. Each subplot corresponds to a specific linear layer, with blue curves representing the small initialization model ($\gamma = 0.5$) and red curves representing the large initialization model ($\gamma = 0.8$). The abscissa denotes the singular value index, and the ordinate denotes the singular value magnitude on a logarithmic scale.

# E  Learning Rate and Weight Decay Coefficient Affecting Solution Phases

A series of studies have experimentally investigated the impact of large learning rates and significant weight decay on the model's flatness and solution consolidation. In our experiments, we found that high learning rates and weight decay coefficients tend to guide the model towards learning inferential solutions. It is important to note that to avoid training instability caused by high learning rates, we modified the settings described in the main text.

In the main text, we artificially assigned non-inferential solutions to the anchor pair (3, 4) as $\mathcal{M}_{(3,4)}(x) = x - 6$. This non-inferential solution significantly disrupts the learning of single anchor mappings for models with small initialization, leading to instability during training with high learning rates. In fact, if we only consider the model's ability to learn inferential solutions, we can treat both anchor pairs (3, 4) and (4, 3) as unseen anchor pairs. Under this setting, if the model can learn the inferential solutions for (3, 4) and (4, 3), it proves that the model learns single anchor mappings. Otherwise, it learns symmetric or other solutions.

As shown in Fig. 14, we study a 3-layer, 1-head transformer model with an initialization rate of $\gamma = 0.5$. We conducted nine independent experiments with different learning rates and weight decay coefficients. The figure presents the mean accuracy of the model's inferential solutions on the unseen anchor pair (4, 3). This visualization clearly demonstrates that larger learning rates and weight decay lead to higher accuracy in inferential solutions, supporting the hypothesis that these hyperparameters play a crucial role in guiding the model towards different phases.

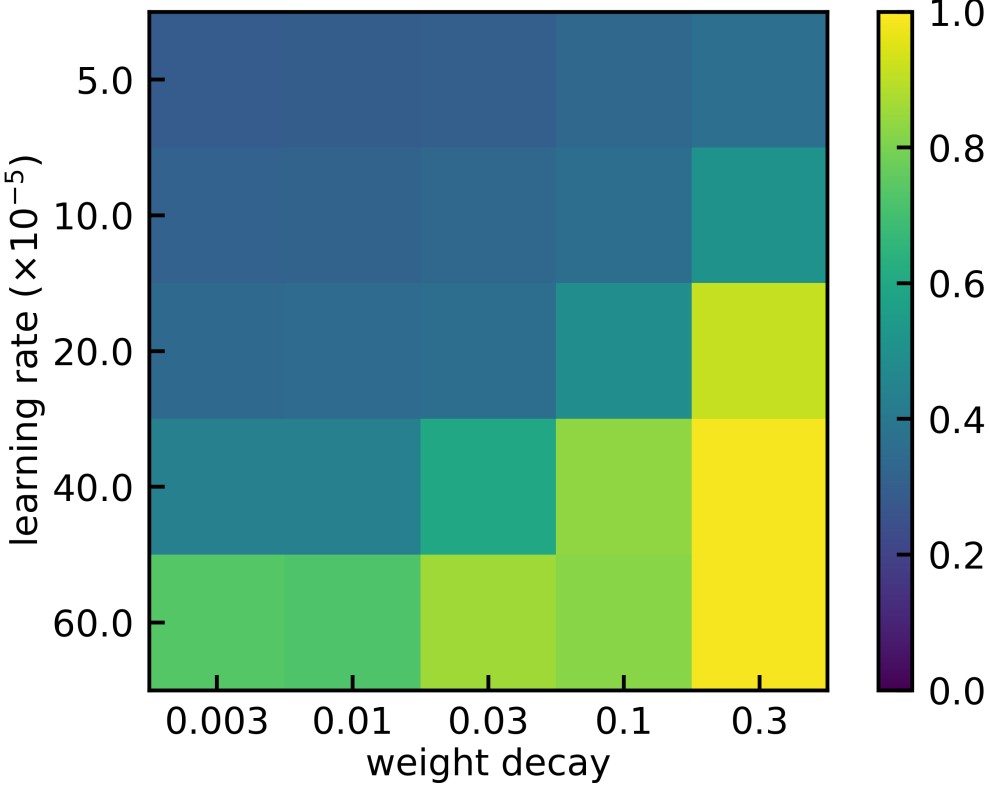

Figure 14: The impact of learning rate and weight decay on the accuracy of inferential solutions in a 3-layer, 1-head transformer model. The heatmap displays the mean accuracy of the model's inferential solutions on the unseen anchor pair (4, 3) across nine independent experiments. The x-axis represents different weight decay coefficients, and the y-axis represents different learning rates. The color bar on the right indicates the accuracy of the inferential solutions on the unseen anchor (4, 3), with higher values corresponding to better performance.

### E.1 Detailed Results for Condensation with Different Learning Rates and Weight Decay Coefficients

In order to show the parameter condensation at different learning rates and different weight decay coefficients, we plotted the cosine similarity of the $W^{Q(1)}$ matrix neurons corresponding to the phase diagram in Fig. 14. As shown in Fig. 15, for each subgraph, we group the weights of two neurons with cosine similarity greater than 0.7 into the same category (adjacent index in the heat map) to highlight the condensation properties of neurons. It is easy to see that as the weight decay coefficient and learning rate increase, the $W^{Q(1)}$ matrix neurons show an obvious condensation trend, implying that the model complexity decreases.

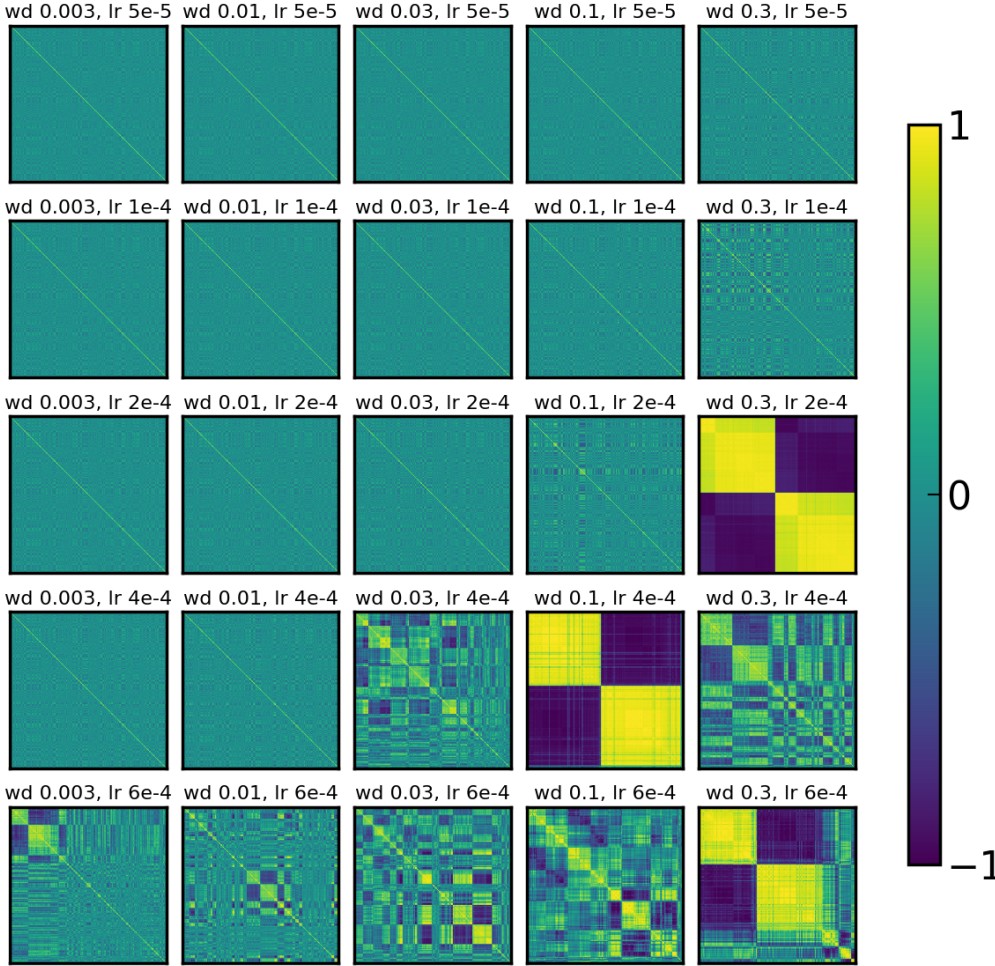

Figure 15: Cosine similarity matrices of $W^{Q(1)}$ neurons at different depths and initialization scales. Each subgraph represents a learning rate and weight decay coefficient. Colors indicate the cosine similarity between neurons, with warmer colors representing higher similarity. The neurons are grouped by cosine similarity greater than 0.7 to highlight the condensation properties.

## E.2 Detailed Results for the Structural Features of the Word Embedding Matrix with Different Learning Rates and Weight Decay Coefficients

We examine the structural changes in the model's word embedding matrix as the learning rate and weight decay coefficient vary. We visualize the parameter matrix using t-SNE. As shown in Fig. 16, as the learning rate and weight decay coefficient increase, the word embedding matrix gradually exhibits distinct structural features. These structural features enable the model to accurately represent the numerical meaning of different items.

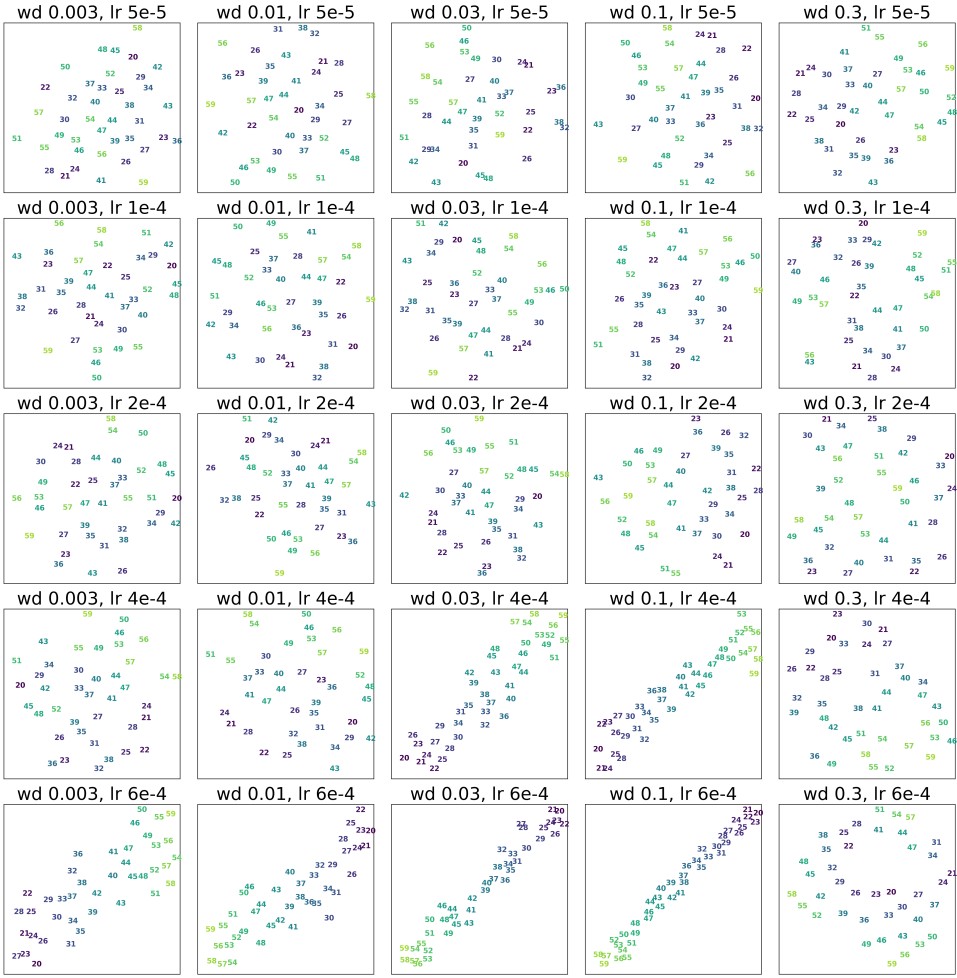

Figure 16: Visualization of the embedding space using t-SNE at different learning rates and weight decay coefficients. Each subgraph represents a different learning rate (from $5 \times 10^{-5}$ to $6 \times 10^{-4}$, top to bottom) and weight decay coefficient (from 0.003 to 0.3, left to right). As the learning rates and weight decay coefficients increase, the word embedding matrix gradually exhibits distinct structural features, indicating decreased model complexity.

## F   Detailed Introduction of Datasets and Additional Results

**Compositional tasks: SCAN and COGS.** SCAN and COGS are two classic compositional tasks, both of which are also synthetic but with more natural language variance, thus being ideal testbeds. For the SCAN dataset, we selected the "Generalizing composition across primitive commands" task, where the "turn left" command only appears in single-command mappings and is trained alongside other composite commands. We assess the model's generalization ability on composite commands that include the "turn left" command. For the COGS dataset, we evaluate the model's in-distribution and out-of-distribution generalization performance after training on the same training set. In-distribution generalization is tested with data constructed from different primitives in the same combinatorial patterns. Out-of-distribution generalization is tested with data that follows different combinatorial rules (the original paper outlines 21 methods for generating out-of-distribution data, from which we generate test data with equal probability).

**Reasoning tasks: PrOntoQA.** PrOntoQA is a synthetic multi-step reasoning dataset where each data point assigns hierarchical relationships among objects and requires the model to determine whether a multi-step reasoning chain is correct. Although we use next-token prediction for training, during testing, we only evaluate the model's accuracy in judging hierarchical relationships (thus, the model's random guessing accuracy is 50%).

**Realistic tasks: Addition task and SlimPajama dataset.** Unlike traditional addition tasks, we use a case-based reasoning intervention experiment (Hu et al., 2024) to study the generalization of rule learning in the addition task. Specifically, we consider the setup: $a + b = c$, where $a, b \in [0, 999]$. We use $a, b \in [400, 600]$ as the test set and the remaining data as the training set. This construction prevents the model from simply mimicking training data similar to the test set. We trained a simple 2-layer 1-head model and a GPT-2 model. As shown in Fig. 17(a), regardless of model size and learning mode, small initialization scales (or large weight decay coefficients) generally lead to good rule generalization, while large initialization scales (or small weight decay coefficients) fail to generalize perfectly.

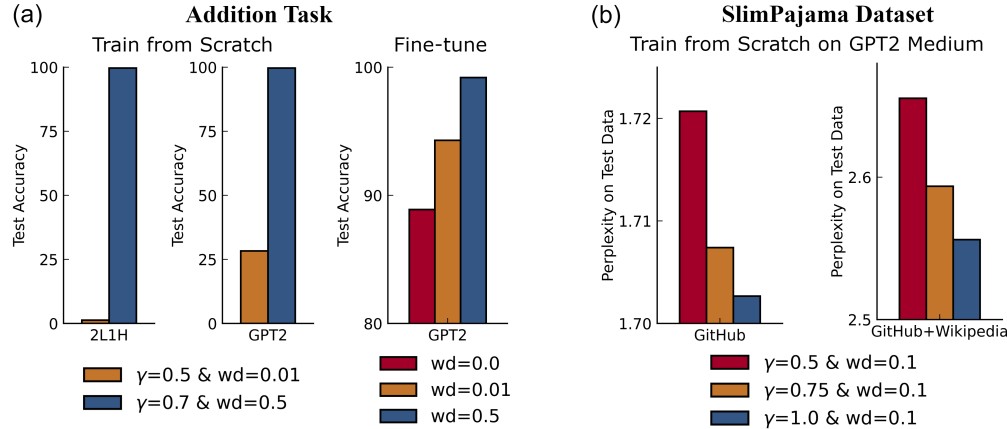

Figure 17: Performance comparison of models with different initialization scales and weight decay coefficients on realistic tasks: Addition task and SlimPajama dataset. (a) In the addition task, we use a case-based reasoning intervention experiment with a test set of $a, b \in [400, 600]$ and the remaining data as the training set. Models with small initialization scales (or large weight decay) generally show better rule generalization. (b) For the SlimPajama dataset, GPT-2 Medium models trained on GitHub and GitHub+Wikipedia data with small initialization scales consistently achieved lower perplexity. The parameters are initialized following a zero-mean normal distribution with a standard deviation of $d_{in}^{-\gamma}$.

For the SlimPajama dataset, we used two data compositions: the GitHub section and the GitHub+Wikipedia section. We trained GPT-2 Medium models with different initializations on both datasets for 40B tokens. As shown in Fig. 17(b), for both data compositions, smaller initialization scales consistently achieved lower perplexity. Notably, by observing the training trajectories, we found that for the GitHub data, the model with small initialization achieved lower perplexity than

the model with large initialization early in training (around 2B tokens). For the GitHub+Wikipedia data, the model with small initialization achieved lower perplexity later in training (around 30B tokens). This further validates the preference of small initialization for reasoning tasks. We will present the detailed training trajectories in the revised manuscript.

