# OpenReview forum: "Initialization is Critical to Whether Transformers Fit Composite Functions by Reasoning or Memorizing"
_NeurIPS.cc/2024/Conference — NeurIPS 2024 poster_

### Official Review · Reviewer_MVAX · 2024-07-07

**Soundness:** 2
**Presentation:** 2
**Contribution:** 2
**Rating:** 6
**Confidence:** 4

**Summary:**

The work mainly investigates whether transformers indeed learn primitive functions or simply learn surficial correlations (i.e., "memorizing") when trained on a mannually-designed toy compositional task, which contains a set of composite numerical functions (e.g., $g_1(x)=x+5, g_2(x)=x+1,...$).

Authors find that the learned mechanism has a phase transition (which correlates with the parameter initialization scale) : smaller initialization scale leads to learning the composite solution ("inferential solution"), where models learn the exact meanings of primitive functions and compose them to solve the task; while larger initialization scale leads to learning solutions based on the surficial correlation in the training data ("symmetric solution").

Centering on this finding, the paper also conducts empirical analysis from the perspectives of information transmission, the similarity of vector representations learned by transformers and model complexity to offer interpretation.

Besides, the paper also finds that there is a positive correlation between the the complexity of the training data (i.e., the number of the data that do not meet the composition rule.) and the difficulty of fitting the training data.

**Strengths:**

1. The topics of this paper, whether the solutions learned by transformer-based models are inferential or just memorizing, are very important and of interest to the NeurIPS community. Besides, the most part of the paper has clear logic structure, thus being easy-to-follow.

2. (Based on the setting discussed in the paper) Authors conduct analysis experiments from multiple aspects, including tracing the information flow within the trained transformers, inspecting the similarity of learned representations and the model complexity, all of which lead to a converging interpretation to the initial observation: smaller initialization scale leads to inferential solutions where larger initialization scale leads to memorizing surficial patterns.

**Weaknesses:**

My main concerns are two-fold:

1. The experiment setting of this work might be over-simplified (as the authors discussed in the limitation part): the synthetic data with little variance (four simple primitive functions, sixteen combinations in total). Though I believe the conclusions in the paper are correct for this setting, I question the generality of the conclusions and the analysis results in other (more complex) settings (especially in this paper, there is only empricial evidence but no explicitly theoretical guarantee for the conclusion).
If we train transformers on other classic compositional tasks (e.g.,: SCAN [1] or COGS [2], both of which are also synthetic but with more natural language variance, thus being ideal testbeds), will we have similar findings?
Besides, the further discussion and validation (at least some initial experiments) on how to use the conclusion to guide the training (fine-tuning) of real pre-trained language models are also lacking. All these factors weaken the contribution of this work.

2. The research question discussed in the paper (i.e., inference v.s. memorization) is highly related to another line of works ("grokking") [3, 4, 5, 6] (i.e., generalization v.s. memorization). The mainly conlusion in the paper (initialization scale correlates to the learned pattern, and even the weight decay part in the Appendix E) is also similar with the conclusion in this "grokking" work [5]. I find the discussion of the relationship (maybe the difference) between this work and the previous "grokking" works are missed in the paper, which weaken the novelty of the work.

[1]: Generalization without systematicity: On the compositional skills of sequence-to-sequence recurrent networks, https://arxiv.org/abs/1711.00350 .

[2]: COGS: A Compositional Generalization Challenge Based on Semantic Interpretation, https://arxiv.org/abs/2010.05465 .

[3]: Grokking: Generalization Beyond Overfitting on Small Algorithmic Datasets, https://arxiv.org/abs/2201.02177 .

[4]: Towards Understanding Grokking: An Effective Theory of Representation Learning, https://arxiv.org/abs/2205.10343 .

[5]: Omnigrok: Grokking Beyond Algorithmic Data, https://arxiv.org/abs/2205.10343 .

[6]: Explaining grokking through circuit efficiency, https://arxiv.org/abs/2309.02390.

**Questions:**

Q1. Regarding the first point in the Weaknesses part:

Q1.1. Can the main conclusion and the analysis results introduced in the paper generalize to other well-studied compositional tasks (e.g., SCAN [1] or COGS [2]) ?

Q1.2. From my point of view, a direct usage for the conclusion introduced in this paper might be initializing the LoRA [3] parameters (or Prompt-Tuning [4] learnable parameters) when fine-tuning pre-trained language models (PLMs). I am wondering if we can observe similar phenomenon when we use PLMs + LoRA/Prompt-Tuning to learn solutions for the task used in this paper or other tasks (e.g., SCAN or COGS)?

Q2. Regarding the second point in the Weaknesses part:

Q2.1. How to understand the difference and the correlation between the conclusions of this work and [5] ?

Q2.2. For larger initialization scale, if we train the transformers far beyond over-fitting, will the learned patterns change (from memorization to inferential solutions)? just as the "grokking" paper described.

Q2.3. What will the learned solution be? if we use larger initialization scale and larger weight decay?

Q2.4. I suggest the authors discuss "grokking" works in the Related Work section.

Q3. An additional question: In Section 6, it is claimed that "Based on our analysis of the mechanisms underlying different solutions, we can predict that models with different initialization scales will exhibit varying degrees of difficulty in learning data with different complexities.". I do not see clear relationship between the initialization scale and data complexity discussed in the previous sections. Could you please explain more about this relationship (between Section 6 and previous sections)? Besides, I think instead of using "epochs for 100% training acc", it is more meaningful to use the "epochs for 100% testing acc" (e.g., using inferential solution as the ground-truth).

Q4. A minor suggestion: the abbreviations are supposed to be introduced when they are used in the paper first time. (e.g., Eq(6), FNN, by the way, I think it should be FFN (Feed-Forward Network) here.)

[1]: Generalization without systematicity: On the compositional skills of sequence-to-sequence recurrent networks, https://arxiv.org/abs/1711.00350 .

[2]: COGS: A Compositional Generalization Challenge Based on Semantic Interpretation, https://arxiv.org/abs/2010.05465 .

[3]: LoRA: Low-Rank Adaptation of Large Language Models, https://arxiv.org/abs/2106.09685

[4]: The Power of Scale for Parameter-Efficient Prompt Tuning, https://arxiv.org/abs/2104.08691

[5]: Omnigrok: Grokking Beyond Algorithmic Data, https://arxiv.org/abs/2205.10343 .

**Limitations:**

The authors do discuss the limitations of this work in Section 7.

---

> ### Author Rebuttal · Authors · 2024-08-06
>
> $\textbf{Weakness 1.1 and Q 1.1}$
>
> The experiment setting of this work might be over-simplified...
>
> $\textbf{Reply}$
>
> We appreciate you suggesting of the SCAN and COGS datasets. We have conducted experiments on these two tasks, and the results are consistent with our conclusions. Additionally, we have tested our findings on a broader range of more complex tasks. For detailed experimental results, please refer to the global response and Reviewer 6WHp's Point 2.
>
> $\textbf{Weakness 1.2}$
>
> ...how to use the conclusion to guide the training...
>
> $\textbf{Reply}$
>
> The primary goal of this work is to guide model architecture design and pre-training processes, rather than fine-tuning on pre-trained models. One direct application is to use different initialization scales and weight decay coefficients based on the type of training data, such as text versus code, to adjust the model's preference for memorization or reasoning.
>
> For example, in the PrOntoQA task, models with small initialization scales exhibit faster learning and better generalization than those with large scales. Similarly, with the SlimPajama dataset, small initialization scales show lower perplexity in the early stages of training on GitHub data (reasoning tasks) compared to large scales. However, when training on combined GitHub and Wikipedia data (reasoning + memorization), small scales show lower perplexity only in the mid-to-late stages (around 120,000 iterations). This aligns with our prediction about initialization scale preferences for different data types.
>
> Furthermore, we can explore new model architectures, such as MoE, with varying initialization scales to balance memory and reasoning preferences, aiming for good generalization and training efficiency.
>
> $\textbf{Q1.2}$
>
> A direct usage … initializing the LoRA.
>
> $\textbf{Reply}$
>
> The low complexity brought by small initialization can indeed guide model fine-tuning. However, LoRA's low-rank design and zero initialization already confer low complexity.
>
> To illustrate, we compared LoRA (r=16, zero initialization for matrix A) with higher complexity learnable parameters (r=768, non-zero initialization for matrix A) on the COGS dataset using a pre-trained GPT-2 model. The LoRA setup achieved 90.4% OOD accuracy, while the higher complexity setup, despite having more learnable parameters, only reached 65.8% OOD accuracy. This supports our understanding of the benefits of small initialization for reasoning tasks.
>
> The limitation of LoRA lies in its rank cap, which prevents it from organically adjusting complexity for more intricate tasks. In contrast, small initialization parameter matrices can flexibly adjust complexity based on task requirements.
>
> $\textbf{Q2.1 and Q2.4}$
>
> The difference and the correlation between this work and [5].
>
> $\textbf{Reply}$
>
> Thank you for highlighting relevant related work. Both our study and [5] explore how parameter initialization scale and weight decay affect model behavior, but with different focuses. We will incorporate a detailed discussion of these points and other relevant grokking studies into the related work section of our revised manuscript.
>
> In [5], the authors vary initialization scale and weight decay to study grokking, focusing on in-distribution generalization. They use unusually large initialization weights, referred to as inducing grokking. Normal and smaller initialization weights do not lead to grokking, so [5] does not provide strong support for generalization differences between normal (what we refer to as large) and small initialization scales.
>
> Our work, in contrast, emphasizes compositional and reasoning tasks, focusing on out-of-distribution generalization. This type of generalization is not a trivial extension of in-distribution generalization. We classify solutions from different initializations into three regimes: symmetric solutions, inferential solutions, and solutions without data generalization. Solutions without data generalization align with large initialization settings in [5] on datasets like MNIST. Both symmetric and inferential solutions show good in-distribution generalization, but [5] does not distinguish between them. We emphasize that for reasoning and compositional tasks with high logical complexity, small initialization is more likely to learn intrinsic data patterns, leading to inferential solutions.
>
> $\textbf{Q2.2.}$
>
> The pattern when far beyond over-fitting.
>
> $\textbf{Reply}$
>
> While this hypothesis is intuitively plausible, we did not observe this phenomenon in our experiments. With large weight decay, the model quickly learns inferential solutions. However, with small weight decay, even after extending the training steps by a factor of 100, the model did not exhibit grokking.
>
> $\textbf{Q2.3}$
>
> What will the learned solution be if we use larger initialization scale and larger weight decay?
>
> $\textbf{Reply}$
>
> If the weight decay is sufficiently large and the training process is stable, the model will learn inferential solutions.
>
> $\textbf{Q3}$
>
> Clarify the relationship between initialization scale and data complexity
>
> $\textbf{Reply}$
>
> We add the following at the beginning of Section 6: "The symmetric solution (large initialization) focuses on symmetric target mappings, while the inferential solution (small initialization) emphasizes logical consistency. Consequently, large initialization models see symmetric mappings as equally complex, whereas small initialization models find non-inferential data more challenging due to the need to understand underlying logical relationships.."
>
> We will use the test set on seen anchors to determine fit quality. Since we have deliberately disrupted the inferential properties of symmetric solutions in multiple datasets, using inferential solutions as the ground truth for testing accuracy is not ideal.
>
> $\textbf{Q4}$
>
> The abbreviations should be introduced.
>
> $\textbf{Reply}$
>
> We will ensure that all abbreviations are introduced upon their first usage in the paper.

---

> > ### Comment · Reviewer_MVAX · 2024-08-08
> > **Thanks for the authors' response**
> >
> > Dear Authors,
> >
> > Thanks a lot for the detailed response.
> >
> > As for the new results on SCAN and COGS, I appreciate the results which demonstrate that initialization scales to some extent affect the generalization performance on unseen (out-of-distribution) compositions. I still have questions regarding this point: Do the model's o.o.d. performance exhibit the phase change or gradual increase with the changing of initialization scale? Moreover, I value the mechanistic insights of two phases discussed in the paper a lot. Could you please validate these findings (i.e., Figure 3 and Figure 4) on SCAN (or COGS)?
> >
> > I will keep thinking about the rest parts in the rebuttal. I write this comment to give you feedback ASAP.
> >
> > Reviewer MVAX.

---

> > > ### Author Response · Authors · 2024-08-08
> > > **Thanks for the reviewers' response**
> > >
> > > Dear Reviewer MVAX,
> > >
> > > Thank you for your appreciation of our results on the SCAN and COGS datasets and for your insightful questions. We immediately conducted further experiments to address your concerns. Here are our responses:
> > >
> > > $\textbf{Point 1}$
> > >
> > > Do the model's o.o.d. performance exhibit a phase change or gradual increase with the changing of initialization scale?
> > >
> > > $\textbf{Reply}$
> > >
> > > Due to time constraints, we re-ran experiments on SCAN and COGS with $\gamma=0.3$ and $\gamma=0.7$, using $\gamma=0.5$ as a reference point. Preliminary results indicate that $\gamma=0.3$ does not lead to a complete absence of OOD generalization (the model still shows some performance on OOD data), which might initially suggest the absence of a phase change. However, detailed analysis revealed that $\gamma=0.3$ models tend to generalize to only a few specific types of OOD data while failing to generalize to others. Here is a detailed explanation:
> > >
> > > In simple terms, OOD data can be subdivided into various types, each with nearly identical generalization difficulty within its type but different difficulties across types. For example, in the COGS dataset, OOD data is divided into 21 categories, each constructed differently and with varying difficulties. If we closely observe the generalization accuracy of each category concerning the initialization scale, we can see a clear phase change.
> > >
> > > For the SCAN dataset, with a training set of 5,000 data points, the attached PDF shows that the $\gamma=0.7$ model generalizes well on OOD data, achieving an accuracy of 23% on OOD data for the $\gamma=0.3$ model. Specifically, if the OOD data consists solely of commands like "I_TURN_LEFT" (e.g., "turn left twice"), the model can achieve nearly 100% accuracy. However, if the output commands include adjacent "I_TURN_LEFT" and "I_TURN_RIGHT," the model almost always fails. Thus, considering OOD data with adjacent "I_TURN_LEFT" and "I_TURN_RIGHT" commands, a phase change appears.
> > >
> > > For the COGS dataset, trained with 20,000 data points, the attached PDF shows that the $\gamma=0.7$ model achieves 65% accuracy on OOD data, while the $\gamma=0.3$ model achieves only 6%, primarily driven by the "pp_dative_to_do_dative" OOD data type. Excluding this specific type, a clear phase change can be observed.
> > >
> > > In summary, for OOD data with uniform difficulty, the phase change in generalization accuracy concerning the initialization scale is evident. However, due to varying difficulties among different OOD data types, the phase change position may slightly vary. This variance can blur the phase change when averaged over multiple data types, creating an appearance of a gradual increase.
> > >
> > > We also recommend examining Figure 2 in the ONE-PAGE PDF. Here, the test data difficulty is uniform (with inference chains of length 3), making the phase change more apparent. As expected, at a data size of 10,000, the large initialization accuracy is only 50% (random guess), while the small initialization accuracy is nearly 100%.
> > >
> > > $\textbf{Point 2}$
> > >
> > > Could you please validate these findings (i.e., Figure 3 and Figure 4) on SCAN (or COGS)?
> > >
> > > $\textbf{Reply}$
> > >
> > > Thank you for appreciating our mechanistic insights. We studied the attention matrices in SCAN. For the $\gamma=0.7$ model, we found that later machine instructions tend to have higher attention weights to earlier corresponding human instructions. Different attention heads in the same layer often focus on different instructions within the same sentence. We observed that shallow attention heads usually focus on a single instruction, while deeper heads associate multiple instructions. Some heads only attend to tokens a few positions ahead. Combining the information of the two types of heads, the machine instruction part can intuitively predict the next instruction. We believe that transforming human language into machine instructions may require the help of MLP layers. For the $\gamma=0.3$ model, the attention matrices do not show clear structures, suggesting that the high regularity of attention matrices is crucial for the model's generalization in compositional tasks.
> > >
> > > Additionally, we examined the generalization mechanisms on PrOntoQA. As the layers deepen, the model's attention matrices transmit information about the next object along the inference chain to the last position in the sequence (the output True or False position). This step-by-step reasoning enables the model to judge the correctness of a Question and output True or False. Similarly, the large initialization model's attention matrices lack such clear structures, resulting in random guessing.
> > >
> > > We again thank the reviewer for the prompt and detailed feedback and hope our rebuttal meets your satisfaction.
> > >
> > > Sincerely,
> > >
> > > The Authors

---

> > > > ### Comment · Reviewer_MVAX · 2024-08-09
> > > > **Thanks for the response.**
> > > >
> > > > Dear Authors,
> > > >
> > > > Thanks for the additional results and analysis. My concerns around the (Weaknesses@1 and Q@1) have now been cleared up. I think it is good to discuss the new experiment settings and results in the revised version of the paper.
> > > >
> > > > Regarding Q3, I would like to clarify my question: since you want to connect the proposed metric "inferential complexity" with the degrees of the difficulty of learning data (I think here we both focus on the solution that models learned, memorizing or inferential). training acc can not unveil which type of solution models learned (both memorizing and inferential can lead to high acc). Hence using testing acc (or a validation acc with o.o.d. data) might be more reasonable.
> > > >
> > > > If the inferential complexity positively correlates with "epochs for achieve a high validation/testing acc"?
> > > >
> > > > Reviewer MVAX.

---

> > > > > ### Author Response · Authors · 2024-08-09
> > > > >
> > > > > Dear Reviewer MVAX,
> > > > >
> > > > > Thank you for your recognition of our new experimental work, and we are pleased to have addressed your concerns. Below, I will explain the rationale behind our design of Figure 6 (which relates to your Q3) and hope to discuss the presentation of this experiment with you.
> > > > >
> > > > > $\textbf{Point 1}$
> > > > >
> > > > >  training acc can not unveil which type of solution models learned (both memorizing and inferential can lead to high acc). Hence using testing acc (or a validation acc with o.o.d. data) might be more reasonable.
> > > > >
> > > > > $\textbf{Reply}$
> > > > >
> > > > > The primary intent behind Figure 6 was to illustrate the difficulty of fitting data with varying inferential complexity using models with different initializations, specifically in terms of the time required for training or generalization on i.d. data. In practice, when there is sufficient data, training accuracy often can reflect test accuracy, and the time required for a model to fit the data becomes an important metric for researchers. We aimed to demonstrate that the same data, when fitted by models with different initializations, can exhibit significantly different fitting speeds, which naturally affects the model's performance under a fixed number of training steps.
> > > > >
> > > > > In essence, the core message we intended to convey with fitting speed aligns with the display method you suggested. When a model is initialized with small values, it tends to learn patterns. Therefore, for data with low inferential complexity, the model achieves a faster fitting speed and learns the inferential solution. Conversely, for data with high inferential complexity, the model's fitting speed decreases, and it is more challenging to learn the inferential solution. On the other hand, for models with large initializations, because the data of varying inferential complexities all disrupt the symmetry in the data, the learning speed across different complexities remains the same, resulting in the model learning a symmetrical solution.
> > > > >
> > > > > Why didn't we initially use the method you suggested (i.e., displaying inferential accuracy) to present our results? There are two main reasons: First, for data with very high inferential complexity, we altered multiple composite mappings in the training data to non-inferential mappings. This made it difficult to define a universally accepted form of inferential mapping under these settings. In an extreme scenario, if all mappings were randomly disrupted into non-inferential mappings, it would be meaningless to compare unseen mappings with the original inferential mappings. The second reason is that we believe the earlier sections of the paper already provide readers with the conclusion that small initialization tends to learn inferential solutions for low-complexity data, while large initialization learns symmetrical solutions for higher-complexity data. Thus, further characterization of data with higher inferential complexity might not add significant value.
> > > > >
> > > > > We acknowledge that your suggestion may address concerns shared by other readers. One possible revision could be to move all the content of Section 6 and the above discussion to the appendix to avoid potential misunderstandings from readers. We would then include the newly added experiments in the main text for a thorough discussion, highlighting the practical significance of small initialization and similar setups in real-world problems. The results of the acc on the inference or symmetry solution are as follows:
> > > > >
> > > > > For inferential complexities of 0 and 1, models with small initialization can learn the inferential solution (inferential complexity 1 is similar to the experimental setup in the main text). For inferential complexity 5, models with small initialization fail to learn either the inferential or symmetrical solution because the form of the inferential solution on unseen anchors becomes unclear based on seen anchors. For models with large initialization, they can learn the symmetrical solution across all inferential complexities, as the symmetry in the data was not disrupted.
> > > > >
> > > > > $\textbf{Point 2}$
> > > > >
> > > > > If the inferential complexity positively correlates with ...
> > > > >
> > > > > $\textbf{Reply}$
> > > > >
> > > > > If we focus on i.d. generalization, the answer is yes for small initializations. In fact, the test accuracy on i.d. data follows a similar trend to the training accuracy, which is the expected outcome we mentioned in our first rebuttal. However, for o.o.d. data, as mentioned above, there may be cases where neither the inferential nor the symmetrical solution is learned. In these cases, the situation does not improve with additional training steps, making the definition of "epochs to achieve a high validation/testing acc" challenging. If we consider "epochs to achieve a high validation/testing acc" as inf if the model fails to achieve perfect generalization on o.o.d. data, then this conclusion would also hold true.
> > > > >
> > > > > We hope this clarifies our approach and look forward to your feedback.
> > > > >
> > > > > Sincerely,
> > > > >
> > > > > The Authors

---

> > > > > > ### Comment · Reviewer_MVAX · 2024-08-10
> > > > > > **I update my score.**
> > > > > >
> > > > > > Dear Authors,
> > > > > >
> > > > > > As most of my concerns are addressed in the discussion phase, I would like to update my score to this paper (from 4 to 6). Please update the new experiment results (provided in the rebuttal phase) to the revised version of the paper.
> > > > > >
> > > > > > Reviewer MVAX.

---

> > > > > > > ### Author Response · Authors · 2024-08-11
> > > > > > >
> > > > > > > Dear Reviewer MVAX,
> > > > > > >
> > > > > > > Thank you very much for your feedback and for taking the time to reconsider your score based on our responses during the discussion phase. We appreciate your constructive comments and are pleased that our additional experiment results have addressed your concerns.
> > > > > > >
> > > > > > > We will update the revised version of the paper with the new experiment results as requested.
> > > > > > >
> > > > > > > Authors.

---

### Official Review · Reviewer_LUj2 · 2024-07-10

**Soundness:** 3
**Presentation:** 2
**Contribution:** 3
**Rating:** 6
**Confidence:** 4

**Summary:**

This paper studies the role of transformer initialization scale on its ability to compositionally generalize. Using a simple arithmetic task, the authors find that when initialization scale is low, the model learns inferential (or compositional) solutions, whereas when the initialization scale is high, models tend to learn memorized solutions, leading to poorer generalization performance. These core findings are supplemented with additional analysis of their model, e.g., through mapping the information flow of each task via analysis of the attention maps. They find that small scale initializations learn token embeddings that are interpretable; that is, the token embeddings generally learn the correct ordinal structure. Finally, they show an analysis that demonstrates that the learning time scales with the inferential complexity of the dataset/task.

**Strengths:**

The study pairs an interesting set of subfields in deep learning: compositionality with neural tangent kernel learning theory. The results are unambiguous and demonstrate strong effects. I have little doubt that these effects are replicable, and that insights from this line of work can be transferred to other domains.

**Weaknesses:**

The primary weaknesses:

1) Other studies demonstrate that this type of arithmetic generalization can be achieved without specific scale-dependent initializations on the arguably more difficult task of length generalization by manipulating positional encodings (Zhou et al., 2023; Zhou et al., 2024; Ruoss et al., 2024; McLeish et al., 2024). In some ways, this paper addresses a simpler form of compositional generalization – systematic compositional generalization – which focuses on generalization on novel combinations of previously seen elements (rather than novel structures of previously seen elements (productive generalization)) (Hupkes et al., 2020).

2) Clarity: I found the paper difficult to understand in some parts. This was partially due to writing, but also partially due to lack of labeling (e.g., figures). For example, the paper often uses the token ID of the anchor pairs (e.g., 3 and 4), rather than the actual referent values of those anchors (-2 and -8), which makes it difficult to follow exactly what figures are showing. Unfortunately, I did not understand the motivation, results, and conclusions of the analyses presented in Section 5.2 and Figure 4. It could be that it was already redundant with results from Section 5.1 (which I did find interesting and easy to follow).

**Questions:**

1) Is there an error in Fig. 1a? It appears the anchor for ‘4’ says ‘8’, when I think it should read ‘-8’.
2) It would be helpful to attach some of the references included in related work to the Introduction section, where they reference the NTK work as guiding the rationale and hypotheses.
3) Is a non-inferenetial composite anchor considered to not follow the traditional rules of arithmetic? That is, [3|4] -> -6, even though $-2 - 8 \neq -6$? The terminology of non-inferential composite anchors was a bit confusing to me. Relatedly, in the dataset, how many non-inferential composite anchors are given? Just 1?
4) Are key items sampled from 20-99 uniformly? Is there a motivation for why this specific set of keys? What are the noise items? The examples provided didn’t seem to provide any examples with noise.
5) In the transformer model initialization, are *all* parameters initialized from the same scale distribution, including: embedding matrix, PE vector, Q/K/V matrices, and MLPs? Or is it a subset of these parameters?
6) I would be curious to see the learning trajectories for Fig. 2c to get a sense of convergence rate across initializations.
7) In general, it would be helpful if anchor pairs were labeled (perhaps with parentheses) of the actual values the correspond to. For example, it is difficult to verify that Fig. 3a/c are doing correctly without having first memorized what the values the anchor pairs refer to.
8) I had trouble understand the motivation and corresponding results for section 5.2, and what the general conclusion/takeaway was. I found the Figure (fig 4) confusing to understand. This is likely related to the above question 7, where it is hard to keep track what the anchor values refer to without having to refer back to Fig 1.
9. Please label the rows and columns of Fig. 5a.
10. Condensation (line 262) is never defined.
11. Regarding Fig 6. There appears to be an effect that adding layers in the small initialization regime improves the sample efficiency of the model. Is this effect true?
12. Lines 132, 133: From the definitions it wasn’t immediately clear what generalization on data meant. Does this mean different key values were used on generalization? Whereas task generalization is specific to anchor pairs?

**Limitations:**

The dataset/task is limited and focuses only on a specific type of ‘systematic compositional generalization’ (i.e., novel combinations of anchor pairs), and does not fully address other forms of compositional generalization, such as length generalization.

Also, the clarity of the paper is a bit difficult to follow, which could potentially limit its impact.

---

> ### Author Rebuttal · Authors · 2024-08-06
>
> $\textbf{Weakness 1}$
>
> Other studies demonstrate that this type of arithmetic generalization can be achieved...
>
> $\textbf{Reply}$
>
> Arithmetic generalization is a widely studied problem, and several works have improved performance primarily on arithmetic tasks by manipulating input token forms or positional encodings based on arithmetic properties. These approaches, however, are often task-specific and do not generalize to broader reasoning tasks. We will add a discussion of these relevant studies to the related work section.
>
> Our approach, on the other hand, aims to provide a more general solution. The discussion on initialization scale in our study is not confined to a specific arithmetic task. The synthetic data serves as an example to illustrate the impact of initialization scale. Our goal is to find a general method that enhances model performance across a wide range of compositional and inferential tasks, not limited to arithmetic tasks.
>
> $\textbf{Weakness 2 and Problem 7}$
>
> It would be helpful if anchor pairs were labeled...
>
> $\textbf{Reply}$
>
> Thank you for this suggestion. We will update the figure captions to label anchor pairs with their corresponding values.
>
> $\textbf{Weakness 2 and Problem 8}$
>
> I had trouble understanding the motivation for Section 5.2...
>
> $\textbf{Reply}$
>
> Section 5.1 demonstrates how models with different initialization scales learn anchor pairs differently. However, this alone does not fully explain the models' behavior when key items are introduced.
>
> Section 5.2 extends this by showing that with the addition of key items, models with small initialization scales can reduce model complexity through structured vector representations. Sequences with the same target have consistent internal representations, requiring the model to learn only around 100 vector representations. In contrast, models with large initialization scales treat each anchor-key item combination as an independent mapping, necessitating the memorization of 800 vector representations (10 types of symmetric anchor pairs and 80 types of key items).
>
> $\textbf{Limitation}$
>
> The dataset/task is limited...
>
> $\textbf{Reply}$
>
> We have also validated our conclusions on a wider range of more complex and real-world datasets, supporting the generality of our findings. For detailed experimental results, please refer to the global response and Reviewer 6WHp's Point 2.
>
> $\textbf{Problem 1}$
>
> Is there an error in Fig. 1a...
>
> $\textbf{Reply}$
>
> We have carefully examined Figure 1a and did not find any instance where the anchor for '4' is mislabeled as '8'. We would appreciate if you could specify the exact location.
>
> $\textbf{Problem 2}$
>
> It would be helpful to attach some of the reference...
>
> $\textbf{Reply}$
>
> Thank you for the suggestion. we add the following sentence to the last paragraph of the Introduction:
>
> "This finding corresponds to a series of studies on different initialization regimes in fully-connected networks, such as NTK [Refs] and mean-field [Refs] regimes."
>
> $\textbf{Problem 3-5}$
>
> Clarification of the experimental setup.
>
> $\textbf{Reply}$
>
> A non-inferential composite anchor does not follow traditional rules. Our intention is to distinguish between symmetric and inferential solutions; otherwise, inferential solutions would be a subset of symmetric solutions under our data setup. There is only one non-inferential composite anchor, as noted in Fig. 1(a).
>
> Key items are sampled uniformly. Since input vectors are encoded in a one-hot format, the key items set only needs to be distinct from the anchors, allowing flexibility in their range. Noise items (gray numbers in "Input data examples" in Fig. 1a) are non-anchor items preceding the anchor, with remaining items as noise. Noise item values are irrelevant to the sequence target.
>
> All parameters are initialized from the same scale distribution.
>
> $\textbf{Problem 6}$
>
> I would be curious to see the learning trajectories...
>
> $\textbf{Reply}$
>
> Due to space limitations in the one-page PDF, we will present the convergence rates for Fig. 2c in the revised manuscript. The one-page PDF includes the convergence rates results of the GPT-2 model on PrOntoQA, a multi-step reasoning dataset, which show similar conclusions to Fig. 2c.
>
> For models with large initialization, the convergence rate initially slows down and then speeds up as the data size increases, demonstrating the phase transition from memorization to inference. In contrast, models with small initialization exhibit consistently fast convergence rates and high generalization accuracy due to their stronger preference for inferential solutions.
>
> $\textbf{Problem 9}$
>
> Please label the rows and columns of Fig. 5a.
>
> $\textbf{Reply}$
>
> We label the rows and columns of Fig. 5a.
>
> $\textbf{Problem 10}$
>
> Condensation (line 262) is never defined.
>
> $\textbf{Reply}$
>
> We add the following explanation to Section 5.3:
>
> ”In small initialization FNNs, the input weights of different neurons cluster along a few isolated orientations, which is referred to as condensation.“
>
> $\textbf{Problem 11}$
>
> ...small initialization regime improves the sample efficiency of the model?
>
> $\textbf{Reply}$
>
> In Fig. 6, we examine the iterations needed for models of varying depths to fit data with different inferential complexities. For high inferential complexity data, some solutions are made non-inferential, requiring higher model complexity and more fitting steps. Increasing layers enhances model complexity, reducing the number of required steps. Since we did not consider generalization in this context, sample efficiency is not directly reflected in the figure.
>
> $\textbf{Problem 12}$
>
> Lines 132, 133: ... what generalization on data meant.
>
> $\textbf{Reply}$
>
> Generalization on data refers to the model's performance on test data where the anchor pairs are seen during training, but the key item and noise items are different. Task generalization, on the other hand, refers to the model's performance on test data with unseen anchor pairs.

---

> > ### Comment · Reviewer_LUj2 · 2024-08-09
> > **Reviewer response to  author rebuttal**
> >
> > I thank the authors for their response to my questions.
> >
> > I think their additional experiments are a good addition to the study, and demonstrate robustness of their results for compositional generalization.
> >
> > A quick clarifying question on Fig 1 of the PDF: How is dataset size varied in the SCAN dataset? I was under the impression the dataset size is fixed (see the SCAN github).

---

> > > ### Author Response · Authors · 2024-08-09
> > > **Thanks for the reviewer's response**
> > >
> > > Dear Reviewer LUj2,
> > >
> > > Thank you for your appreciation of our additional experimental work, and we're glad to have addressed your concerns. Below, I will explain how we constructed our dataset:
> > >
> > > $\textbf{Point 1}$
> > >
> > > How is dataset size varied in the SCAN dataset? I was under the impression the dataset size is fixed (see the SCAN github).
> > >
> > > $\textbf{Reply}$
> > >
> > > The official dataset size provided by the authors is indeed fixed. For our SCAN dataset construction (as well as for other datasets like COGS), we based our work on the dataset provided by the original authors. To better illustrate the significant differences in performance between models with different initializations on varying dataset sizes, we conducted experiments using different dataset sizes. The smaller datasets were all derived by subsetting larger ones. For example, the 1w-sized dataset was derived from the full dataset available on the original authors’ GitHub, and the 0.5w-sized dataset was further subset from the 1w-sized dataset, and so on. Additionally, the datasets used by different initialization models with the same dataset size are identical. This approach allows us to observe the relationship between model performance and dataset size while maintaining the official data distribution.
> > >
> > > We hope our rebuttal satisfies your concerns. Please feel free to reach out with any further questions, and we look forward to your feedback.
> > >
> > > Sincerely,
> > >
> > > The Authors

---

> > > > ### Comment · Reviewer_LUj2 · 2024-08-12
> > > > **Thanks**
> > > >
> > > > I thank the author for their clarification -- the approach makes sense.
> > > >
> > > > I keep my score at a 6, due to existing concerns regarding the paper's clarity and presentation. Nevertheless, I think the results are sound and interesting.

---

> > > > > ### Author Response · Authors · 2024-08-13
> > > > >
> > > > > Dear Reviewer LUj2,
> > > > >
> > > > > Thank you for your thoughtful response and for recognizing the soundness and interest of our work. We appreciate your feedback and understand the concerns regarding the paper's clarity and presentation. As mentioned in our rebuttal, we will be making further revisions to the manuscript based on your suggestions and those of the other reviewers.
> > > > >
> > > > > The Authors.

---

### Official Review · Reviewer_6WHp · 2024-07-12

**Soundness:** 3
**Presentation:** 4
**Contribution:** 3
**Rating:** 6
**Confidence:** 4

**Summary:**

Transformers show remarkable capabilities but struggle with compositional tasks, raising questions about their true understanding versus input-output mapping. The authors investigates transformers' generalization to unseen compositional tasks using anchor functions, revealing that initialization scale impacts solution types. With smaller initialization scales, models tend to infer solutions, while larger scales lead to symmetric solutions. Analysis shows symmetric solutions combine anchor information without compositional understanding, whereas inferential solutions capture underlying primitives. These insights suggest model complexity influences the ability to learn compositional primitives, with initialization scale being crucial.

**Strengths:**

- The investigation into the effect of initialization on compositional abilities of transformers is well-motivated and practically useful.
- The discovery of phase transitions from symmetric memorization to generalization based on initialization scale is compelling.

**Weaknesses:**

- While the problem and findings are new, the approach lacks novelty.
- The focus on symmetric solutions is novel, but the generalizability to real data and a broader range of models remains uncertain.

**Questions:**

- Do you have a solid definition for "Phase Transition"? If so, how did you define it?
- Do you have any additional results with other data/models (no need to run new experiments for the rebuttal)?

**Limitations:**

Experiments were conducted primarily on a simple synthetic setup. Future work should verify the extent to which these findings can be generalized and explore the implications of these findings on real-world applications of transformers.

---

> ### Author Rebuttal · Authors · 2024-08-06
>
> $\textbf{Point 1}$
>
> While the problem and findings are new, the approach lacks novelty.
>
> $\textbf{Reply}$
>
> We appreciate your recognition of the novelty in our problem and findings. While our approach may appear straightforward, we hope reviewers appreciate the challenges we overcame in experimental design and mechanistic study. The relationship between initialization scale and task generalization (OOD generalization) is not well understood before, making the experimental design itself a novel direction. Specifically, in this experiment, both symmetric and inferential solutions satisfy data generalization, meaning both achieve traditional in-distribution generalization. Studying in-distribution and out-of-distribution generalization in a well-analyzed synthetic task is inherently challenging, and it was uncertain whether the model would exhibit task generalization capability.
>
> Furthermore, we analyzed the reasons behind task generalization at the parameter level, highlighting the importance of low effective complexity in compositional (inferential) tasks. Our work paves the way for new directions in developing models that integrate inference and memorization and explaining the emergence of inferential capabilities from a dynamical perspective. These contributions add significant value to our study and provide a foundation for future research in this area.
>
> $\textbf{Point 2}$
>
> The focus on symmetric solutions is novel, but the generalizability to real data and a broader range of models remains uncertain.
>
> $\textbf{Reply}$
>
> We appreciate your recognition of the novelty in our focus on symmetric solutions. Experiments on synthetic data are crucial for discovering symmetric solutions and understanding the underlying mechanisms of transformer models.
>
> To address concerns about generalizability, we conducted additional experiments using various datasets and models. We tested different scales of GPT-2 models on datasets focused on compositional and inferential tasks, including SCAN [1], COGS [2], PrOntoQA [3], and the Addition task using case-based reasoning intervention experiments [4]. We also used real-world data from the GitHub (33B) and Wikipedia (22B) sections of the SlimPajama dataset [5]. The results from these experiments consistently supported our conclusions, demonstrating that the findings about initialization scale and its impact on learning behavior hold true across different tasks and model scales.
>
> We will include these additional results in the revised manuscript to provide stronger evidence for the generalizability of our findings. Detailed experimental results can be found in the global response and our one-page PDF.
>
> $\textbf{Point 3}$
>
> Do you have a solid definition for "Phase Transition"? If so, how did you define it?
>
> $\textbf{Reply}$
>
> The term "Phase Transition" originates from statistical mechanics, where it describes a transformation between different states of matter, when a control parameter, like temperature or pressure, reaches a critical value. This term is also often used in a series of machine learning studies [6].
>
> In the context of our paper, we use "Phase Transition" to describe a similar phenomenon observed in the behavior of transformer models as a function of the initialization scale. Specifically, we define the phase transition as the point at which the model's learned solutions shift from being inferential (capturing the underlying compositional primitives) to symmetric (memorizing mappings without understanding the compositional structure). This shift occurs as the initialization scale crosses a critical threshold ($\gamma=0.5$ in this work), analogous to the critical temperature or pressure in physical phase transitions.
>
> $\textbf{References}$
>
> [1] Generalization without systematicity: On the compositional skills of sequence-to-sequence recurrent networks, ICML 2018.
>
> [2] COGS: A Compositional Generalization Challenge Based on Semantic Interpretation, EMNLP 2020.
>
> [3] Language Models Are Greedy Reasoners: A Systematic Formal Analysis of Chain-of-Thought, ICLR 2023.
>
> [4] Case-Based or Rule-Based: How Do Transformers Do the Math? ICML 2024.
>
> [5] SlimPajama: A 627B token cleaned and deduplicated version of RedPajama. https://www.cerebras.net/blog/slimpajama-a-627b-token-cleaned-and-deduplicated-version-of-redpajama
>
> [6] Phase diagram for two-layer ReLU neural networks at infinite-width limit. Journal of Machine Learning Research (2021).

---

> > ### Comment · Reviewer_6WHp · 2024-08-12
> >
> > Thank you for the detailed response and the additional experiment. I don't have any further questions, and I'll maintain my score toward acceptance.

---

> > > ### Author Response · Authors · 2024-08-13
> > >
> > > Dear Reviewer 6WHp,
> > >
> > > Thank you for your detailed feedback and for taking the time to review our work. We greatly appreciate your recognition of our efforts, and we are pleased that our additional experiments have addressed your concerns.
> > >
> > > The Authors.

---

### Author Rebuttal · Authors · 2024-08-06

We thank the reviewers for your thoughtful and insightful comments. We have addressed every comment, and believe that, taken together, the reviewers' comments have improved the manuscript significantly. To address reviewers' common concerns, we have supplemented relevant experimental results, added more discussion, and improved presentation. Our improvements to the draft mainly include the following points.

$\textbf{Adding Numerical Verification on Other Realistic Tasks.}$

We validated the performance of models with different initialization scales and weight decay settings across a series of compositional and reasoning tasks. Below, we introduce each task and the corresponding results. Notably, models with small initialization (large weight decay) $\textbf{consistently outperformed}$ those with large initialization (small weight decay) across all tasks. For the figures mentioned below, please refer to the PDF of the global response.

$\textbf{1. Compositional tasks: SCAN and COGS.}$ SCAN [1] and COGS [2] are classic compositional tasks with more natural language variance. For SCAN, we used the "Generalizing composition across primitive commands" task, where the "turn left" command appears only in single-command mappings during training. We assess generalization on composite commands including "turn left". For COGS, we evaluate in-distribution and out-of-distribution generalization after training on the same set. In-distribution tests use data with different primitives in the same combinatorial patterns, while out-of-distribution tests use data following different combinatorial rules.

As shown in Fig. 1, we display the generalization performance of models with different initialization scales and weight decay across various data sizes. Small initialization and large weight decay consistently outperform large initialization and small weight decay across different task types and data scales. Notably, in the COGS task, even when the in-distribution generalization of both settings reaches over 99\%, the difference in out-of-distribution generalization remains significant.

$\textbf{2. Reasoning task: PrOntoQA.}$ PrOntoQA [3] is a synthetic multi-step reasoning dataset that assigns hierarchical relationships among objects, requiring the model to determine the correctness of reasoning chains. We use next-token prediction for training but evaluate the model's accuracy in judging hierarchical relationships during testing (random guessing accuracy is 50%).

Fig. 2 shows convergence rates and generalization errors for models with large initialization and small weight decay (Fig. 2(a)) and small initialization and large weight decay (Fig. 2(b)). For large initialization, as data size increases, the convergence rate first decreases due to increased memorization difficulty. However, as the data size grows further, the model, constrained by its complexity, shifts to reasoning, which then improves generalization. In contrast, models with small initialization prefer reasoning from the start, leading to faster convergence and better generalization.

$\textbf{3. Realistic tasks: Addition task and SlimPajama dataset.}$ Unlike traditional addition tasks, we use a case-based reasoning intervention experiment [4] to study the generalization of rule learning in the addition task. Specifically, we consider the setup: $a + b = c$, where $a, b \in [0, 999]$. We use $a, b \in [400, 600]$ as the test set and the remaining data as the training set. This construction prevents the model from simply mimicking training data similar to the test set. As shown in Fig. 3(a), regardless of model size and learning mode, small initialization scales (or large weight decay) generally lead to good rule generalization, while large initialization scales (or small weight decay) fail to generalize perfectly.

For the SlimPajama dataset [5], we used two data compositions: the GitHub section and the GitHub+Wikipedia section. We trained GPT-2 Medium models with different initializations on both datasets for 40B tokens. As shown in Fig. 3(b), for both data compositions, smaller initialization scales consistently achieved lower perplexity. Notably, the model with small initialization achieved lower perplexity earlier in training (around 2B tokens) for GitHub data and later (around 30B tokens) for GitHub+Wikipedia data, validating the preference of small initialization for reasoning tasks. Detailed training trajectories will be presented in the revised manuscript.

$\textbf{Enhancing Clarity and Adding Related Work Discussion }$

To improve clarity and enhance the reading experience, we have made several revisions to the paper:

• Added anchor mappings in figure captions.

• Clarified motivations in Sections 5.2 and 6.

• Included necessary learning trajectories in the appendix.

• Provided definitions for terms such as "condensation".

• Expanded abbreviations like "FFN" to their full forms.

We have also added discussions on relevant works concerning grokking, NTK and arithmetic tasks to highlight our study's contributions. While some existing works explore the impact of initialization on models, our focus on how initialization affects data learning patterns, particularly small initialization's preference for compositional and reasoning tasks, is both novel and significant. Additionally, our exploration of the internal mechanisms of models under different initialization settings provides valuable insights to the field.

$\textbf{References}$

[1] Generalization without systematicity: On the compositional skills of sequence-to-sequence recurrent networks, ICML 2018.

[2] COGS: A Compositional Generalization Challenge Based on Semantic Interpretation, EMNLP 2020.

[3] Language Models Are Greedy Reasoners: A Systematic Formal Analysis of Chain-of-Thought, ICLR 2023.

[4] Case-Based or Rule-Based: How Do Transformers Do the Math? ICML 2024.

[5] SlimPajama: A 627B token cleaned and deduplicated version of RedPajama.

---

### Decision · Program_Chairs · 2024-09-25

**Decision:**

Accept (poster)

**Comment:**

The reviewers' initial assessment was that while the paper introduced worthwhile ideas and a sound method, the empirical validation using "toyish' datasets did not indicate whether the method could generalize to larger-scale compositional tasks (including non-arithmetic ones). The reviewers also made other remarks regarding the possible limited technical novelty of the "methods" introduced in the paper, the lack of clarity of some of the reported results, and possible missing discussion of related works (e.g. related to length generalization in arithmetic problems and grokking).

The authors provided a comprehensive response, which included several additional results on larger-scale datasets, including more "standard" ones (e.g., SCAN for "compositional navigation commands" and COGS "Compositional Generalization Challenge Based on Semantic Interpretation") and also reasoning and language modelling datasets/tasks. The new results match the smaller-scale results from the initial submission. These were also useful in demonstrating the work's aim was more general than arithmetic problems.

Regarding "grokking," the authors clearly explained the similarities and differences and agreed to add that discussion to the paper. Similarly, the authors provided specific clarifications to be added to the paper.

While novelty is important, there are different types of papers, and this submission provides and validates timely novel insights. As such, I find that it stands without a novel method.

Finally, the reviewers agree that this paper should be accepted, and I am happy to recommend acceptance. Congratulations!